# Post-Training Adaptive Conformal Prediction for Incomplete Time Series

**Baiting Chen**                                                              *brantchen@g.ucla.edu*
*Department of Statistics and Data Science*
*UCLA*

**Xiaofan Zhou**                                                                 *xzhou77@uic.edu*
*Department of Computer Science*
*University of Illinois Chicago*

**Lu Cheng**                                                                      *lucheng@uic.edu*
*Department of Computer Science*
*University of Illinois Chicago*

**Reviewed on OpenReview:** *https://openreview.net/forum?id=KMBU4wx79B*

## Abstract

Conformal Prediction (CP) is widely used for uncertainty quantification but faces significant challenges with time series due to non-exchangeability. The issue is exacerbated by missing data, where the exponential growth of missing patterns makes existing approaches computationally expensive and unable to adequately represent each missing pattern. To address this, we propose a novel approach that uses a post-training Neural Network to handle temporal dependencies and structured missingness in time series data. With a novel non-conformity score function, our method improves conditional coverage for different missing patterns, ensuring prediction intervals are both reliable and informative. We introduce features that capture different missingness mechanisms, enabling the model to adapt to various patterns. Theoretically, we establish asymptotic validity for conditional coverage with adaptive adjustments. Experiments on semi-synthetic benchmarks demonstrate the method's efficiency in producing tight prediction intervals while maintaining group conditional coverage.

## 1 Introduction

Conformal Prediction (CP) (Vovk et al., 1999; 2005; Papadopoulos et al., 2002) has emerged as a powerful framework for providing uncertainty quantification in time series forecasting, which is especially important in high-stakes domains (Gibbs & Candes, 2021; Barber et al., 2023; Xu & Xie, 2021; 2023; Zaffran et al., 2022). At the same time, the increasing scale of time series data has led to more frequent occurrences of missing values (Emmanuel et al., 2021; Little & Rubin, 2019). In areas such as healthcare, for instance, clinical measurements are often collected irregularly across features, resulting in substantial missingness (Lipton et al., 2016). Recent advances have begun to explore CP methods that address missing data (Zaffran et al., 2023), introducing the notion of Mask-Conditional-Validity (MCV) to ensure coverage conditional on missing patterns. These developments have provided an important foundation for handling uncertainty in incomplete data settings. Building on these insights, there remains an opportunity to extend CP beyond exchangeable data and the Missing Completely At Random (MCAR) assumption toward more general time series contexts, where temporal dependencies and structured missingness are prevalent. In time series, not only is the exchangeability assumption violated, but the missingness is often correlated with the input features or the underlying data generation process, falling into the categories of missing at random (MAR) or missing not at random (MNAR). Under these settings, standard conformal calibration can become miscalibrated across time and across missingness patterns, producing conservative intervals in stable periods yet undercoverage in

missing-heavy or shifting regimes. This motivates the development of new CP techniques capable of capturing both the *temporal dependencies* and the *structured missingness* inherent in time series, while maintaining reliable uncertainty quantification.

This work aims to quantify uncertainty in time series with missing values, ensuring prediction intervals are both tight, reflecting high prediction efficiency, and achieve reliable conditional coverage across various missing mechanisms. Two major challenges arise in this setting. First, the complexity of missingness increases exponentially as the variety of missing patterns grows, making traditional CP methods for group conditional coverage ineffective especially when available data is sparse, leading to insufficient representation for each missing pattern. Second, there are two types of distribution shifts that further complicate the problem: one stemming from differences in structured missingness between calibration and test data, and another due to the inherent temporal dependencies in time series.

To address these challenges, we developed ACP-TSM, an Adaptive CP algorithm capable of handling three types of Missingness in Time Series: MCAR, MAR, and MNAR. To overcome data scarcity in individual groups, we propose using hand-engineered features that capture the patterns of missingness, allowing us to leverage the entire calibration dataset in the CP process, rather than limiting it to samples with similar missingness to the test instance, as previous methods have done. To mitigate the effects of distribution shifts caused by missingness and temporal dependencies, we incorporate an NN specifically designed for time series data. This NN enhances non-conformity scores by directly minimizing the group conditional coverage gap, using the hand-engineered features to better capture group-level information. Additionally, inspired by our empirical observation that samples from nearby time steps tend to exhibit similar prediction interval lengths, we develop a novel loss function aimed at explicitly reducing the prediction inefficiency loss. Our contributions are summarized as follows:

- To the best of our knowledge, our work is the first to explore CP in the setting of time series data with missing covariates. We address the challenges posed by temporal dependencies and structured missingness, focusing on three common missing data mechanisms: MCAR, MAR, and MNAR.

- We propose a post-training LSTM for CP with dual loss functions for both group conditional coverage and prediction efficiency (i.e., tight prediction intervals). By introducing novel feature engineering techniques and nearby-step context, we allow the LSTM to learn structured patterns of missingness, therefore, enhancing CP performance.

- We provide theoretical and empirical evidence demonstrating that ACP-TSM effectively improves computational efficiency and CP performance across three common missing mechanisms.

## 2 Related Works

**Conformal Prediction** CP, pioneered by Vovk et al. (2005), has become a cornerstone in uncertainty quantification due to its model-free and distribution-free property (Shafer & Vovk, 2008; Zhou et al., 2025). The work in this domain can be summarized of CP into two branches: improve the efficiency of CP (Romano et al., 2020b; Fisch et al., 2020; Yang & Kuchibhotla, 2021; Stutz et al., 2021; Bai et al., 2022; Ndiaye, 2022) and generalize CP to different settings (Romano et al., 2019; Johansson et al., 2013; 2014; Candès et al., 2023; Papadopoulos et al., 2011; Bhatnagar et al., 2023). Comprehensive insights into CP and its theoretical underpinnings are provided by Angelopoulos & Bates (2021). Adaptation of CP for time series can be divided into two primary trends that deviate from the traditional assumption of exchangeability. The first trend focuses on adaptively adjusting the error rate $\alpha$ during the testing phase to enhance coverage accuracy. This approach is demonstrated in works such as Romano et al. (2020a); Zaffran et al. (2022); Lin et al. (2022); Angelopoulos et al. (2024). The second trend (Tibshirani et al., 2019; Xu & Xie, 2021; 2023; Barber et al., 2023; Auer et al., 2023; Chen et al., 2024) emphasizes assigning weights to historical non-conformity scores, giving greater importance to those that reflect the current scenario more accurately. Achieving conditional coverage based on covariates has been shown to be impossible (Foygel Barber et al., 2021; Vovk et al., 2005). Several works have instead focused on developing methods for achieving conditional coverage within predefined groups (Bhattacharyya & Barber, 2024; Feldman et al., 2021; Wang et al., 2024a;

Zhou & Sesia, 2024; Liu & Wu, 2024). The work most closely aligned with ours is Xie et al. (2024), where we focus on specified group conditional coverage, while they target covariate-based conditional coverage.

**Time Series with Missing Values** Time series data is particularly prone to missing values during recording, and prior research has primarily focused on designing imputation models to handle missing values based on the observed data (Pratama et al., 2016; Moritz & Bartz-Beielstein, 2017; Che et al., 2018; Fang & Wang, 2020; Wang et al., 2024b). The current state-of-the-art (SOTA) imputation methods are predominantly deep learning-based (Liu et al., 2023; Cini et al., 2021), which excel in capturing complex temporal dependencies and patterns in missing data. Another line of research examines different missing data mechanisms in time series (Lipton et al., 2016; Kreindler & Lumsden, 2016). To the best of our knowledge, our work is the first to study CP in time series data with missing covariates. The most closely related work is by Zaffran et al. (2023), which proposes *CP-MDA-Exact* and *CP-MDA-Nested* to address missingness for exchangeable data with specifically designed matched calibration data. Our research advances prior research by addressing the missingness of non-exchangeable data. Our core idea is based on the observation that adjacent time steps have more similar CP prediction intervals. Therefore, we propose to boost prediction intervals by employing a post-training LSTM with dual loss functions that explicitly minimize the length of the prediction intervals. In addition, our method incorporates more calibration data without the need to select specific missing pattern data that match the test data. This design allows us to effectively utilize a wider range of available data to provide more efficient and accurate prediction intervals.

## 3 Preliminaries and Problem Setup

**Missing Mechanisms** can be represented using a binary indicator $M \in \{0,1\}^d$, where $M_j = 1$ indicates that the $j$-th feature is missing, and $M_j = 0$, otherwise. We briefly introduce three common missing data mechanisms (Rubin, 1976). Mathematical descriptions are in Appendix B.

- MCAR: Missingness is independent of both observed and unobserved values of the data.
- MAR: Missingness is dependent only on the observed portion of the data.
- MNAR: All scenarios outside MCAR and MAR fall under MNAR. Here, the missingness depends on both the observed and unobserved data.

**Conformal Prediction (CP).** Conformal prediction is an uncertainty quantification methodology with finite-sample statistical guarantees without strong modelling assumptions (Vovk et al., 2005; Angelopoulos & Bates, 2021). CP wraps around any trained model and uses a calibration dataset to output a prediction set (or interval) $C(x)$ that is guaranteed to contain the true outcome with high probability. One of the most common approaches in CP is the *split conformal prediction* method (Papadopoulos et al., 2002), which starts with a trained model and assesses its efficacy on a calibration set of paired examples $\{(X_i, Y_i)\}_{i=1}^n$. Central to split CP is the foundational assumption that the data are exchangeable, ensuring that the errors observed in the test set will be consistent with those from the calibration set. Define a non-conformity score function $s(x, y)$ that quantifies the disagreement between predicted and ground-truth values. The coverage guarantee of CP is defined as follows:

$$P(Y_{n+1} \in C(X_{n+1})) \geq 1 - \alpha, \quad C(X_{n+1}) = \{y : s(X_{n+1}, y) \leq \hat{q}\}. \tag{1}$$

where $(X_{n+1}, Y_{n+1})$ is a new data point, $\hat{q}$ is the $\lceil (1-\alpha)(n+1) \rceil$-th smallest elements among $\{s(X_i, Y_i)\}_{i=1}^n$ and $\alpha$ signifies a predefined miscoverage rate.

**Nonexchangeability.** Real-world settings encounter challenges such as data drift and inter-dependencies, necessitating further adaptations of CP to these non-exchangeable settings. One common solution is to use reweighting (Barber et al., 2023; Auer et al., 2023). For example, given a set of pre-specified weights $\{w_i\}_{i=1}^n$, $w_i \in [0, 1]$, the coverage guarantee for seminal work NexCP (Barber et al., 2023) is articulated as follows:

$$P(Y_{n+1} \in C(X_{n+1})) \geq 1 - \alpha - \sum_{i=1}^n \tilde{w}_i d_{\text{TV}}(Z, Z^i), \quad \tilde{w}_i = \frac{w_i}{1 + \sum_{i=1}^N w_i}, \tag{2}$$

where $Z = (X_1, Y_1), \ldots, (X_n, Y_n), (X_{n+1}, Y_{n+1})$ represents a sequence of $n$ calibration examples along with a subsequent test example and $\tilde{w}_i$ is the normalized weight. The term $Z^i$ denotes the sequence $Z$ after the

$i$-th pair $(X_i, Y_i)$ is swapped with the test example $(X_{n+1}, Y_{n+1})$, and $d_{\text{TV}}(Z, Z^i)$ quantifies the dissimilarity introduced by this swap. To construct prediction sets, NexCP determines the quantile threshold $\hat{q}$ by the following equation:

$$\hat{q} = \inf \left\{ q : \sum_{i=1}^{n} \tilde{w}_i 1\{s_i \leq q\} \geq 1 - \alpha \right\}. \tag{3}$$

*Conformalized Quantile Regression* (CQR) (Romano et al., 2019) is an adaptive version of SCP that fits two quantile regressions $\hat{q}_{\text{low}}$ and $\hat{q}_{\text{upp}}$ on the training data. The non-conformity score is defined as $s(x, y) = \max\left(\hat{q}_{\text{low}}(x) - y, y - \hat{q}_{\text{upp}}(x)\right)$. The prediction interval for a new test point $x$ is then defined as: $C(X_{n+1}) := \{y : s(X_{n+1}, y) \leq \hat{q}\}$.

**Problem Formulation** Let $\{(X^{(t)}, Y^{(t)}, M^{(t)})\}_{t=1}^{T}$ denote a sequence of $T$ observations, where $X^{(t)} \in \mathbb{R}^d$ represents the $d$-dimensional feature vector at time $t$, $Y^{(t)} \in \mathbb{R}$ represents the corresponding scalar outcome or response variable, $M^{(t)} \in \{0, 1\}^d$ is a binary mask vector that indicates missingness in $X^{(t)}$. Let $k \in \{1, \ldots, K\}$ denote the forecasting horizon, where $K \in \mathbb{N}_+$ is a user-specified maximum horizon. The goal is to construct a prediction interval $C_\alpha\left(X^{(T+k)}, M^{(T+k)}\right) = \left[L_{T+k}(\cdot), U_{T+k}(\cdot)\right]$ for $Y^{(T+k)}$ at time $T + k$, given the test data $X^{(T+k)}$ and missingness pattern $M^{(T+k)}$ such that it satisfies both **marginal coverage** and **conditional coverage** w.r.t. different missing patterns, while also achieving **high prediction efficiency**, measured by the (empirical) **average width** of the prediction intervals $\text{Width}(C_\alpha) := \frac{1}{|K|} \sum_{k \in K} \left(U_{T+k} - L_{T+k}\right)$.

**Marginal Coverage:** Given $\alpha \in (0, 1)$, the prediction interval should include the true outcome $Y^{(T+k)}$ with probability at least $1 - \alpha$ for the test data $X^{(T+k)}$:

$$\mathbb{P}\left(Y^{(T+k)} \in C_\alpha(X^{(T+k)}, M^{(T+k)})\right) \geq 1 - \alpha. \tag{4}$$

**Group Conditional Coverage:** For any group $G_m \in \mathcal{G}$, where each group corresponds to a specific missingness pattern $M^{(T+k)} = m$, the conditional coverage guarantee is defined as follows:

$$\mathbb{P}\left(Y^{(T+k)} \in C_\alpha\left(X^{(T+k)}, M^{(T+k)}\right) \,\Big|\, M^{(T+k)} = m\right) \geq 1 - \alpha. \tag{5}$$

Correspondingly, the marginal coverage is an immediate consequence of a group-conditional coverage guarantee over missingness patterns. If we can ensure that for Eq 5 holds for each group, then marginal overage follows directly by the law of total probability:

$$\mathbb{P}\left(Y^{(T+k)} \in C_\alpha\left(X^{(T+k)}, M^{(T+k)}\right)\right) = \sum_{m=1}^{M} \mathbb{P}\left(Y^{(T+k)} \in C_\alpha\left(X^{(T+k)}, M^{(T+k)}\right) \,\Big|\, M^{(T+k)} = m\right) \mathbb{P}\left(M^{(T+k)} = m\right)$$

$$\geq \sum_{m=1}^{M} (1 - \alpha) \mathbb{P}\left(M^{(T+k)} = m\right) = 1 - \alpha. \tag{6}$$

Conditional coverage ensures that prediction intervals are neither overly conservative nor suffer from under-coverage in each subgroup, and helps manage different levels of risk or tendencies associated with varying missingness patterns. For example, consider a time series dataset used for predicting healthcare outcomes, where some subgroups of patients have more missing data due to factors such as inconsistent access to healthcare. If a prediction model systematically produces wider prediction intervals (overly conservative) for these subgroups, the result could be less actionable information, leading to fewer effective treatments. Conversely, if the prediction intervals are too narrow (under-coverage), it may lead to overconfident decisions, increasing the risk of harmful outcomes for vulnerable subgroups.

## 4 Methodology

In this section, we first discuss why a direct combination of conformal prediction for time series and standard missing-covariate handling can fail. We then introduce ACP-TSM in Section 4.2, which uses a dual-objective loss to explicitly target group-conditional coverage and prediction efficiency via missingness-related features,

while preserving the desired marginal coverage guarantee. Our method is CP-based, offering the advantages of being model-agnostic, distribution-free, and lightweight, even when dealing with exponential-growth group sizes due to missing patterns.

## 4.1 A Naive Approach

There are existing techniques that adapt CP for time series data and missing covariates, respectively. A straightforward approach is to adapt CP to each challenge independently and then combine them. For missing covariates, CP-MDA-Exact (Zaffran et al., 2023) builds a *test-specific* calibration set by retaining only those calibration points whose missingness pattern is compatible with the current test point, so that calibration data and test date become exchangeable. Formally, *CP-MDA-Exact* generates an additional calibration set $\text{Cal}^{(\text{test})}$ as $\text{Cal}^{(\text{test})} = \left\{ k \in \text{Cal} \mid M^{(k)} \subseteq M^{(\text{test})} \right\}$, where Cal is the original calibration set, $M^{(k)}$ and $M^{(\text{test})}$ represent the missingness patterns of calibration data point $k$ and the current test point. Given $\text{Cal}^{(\text{test})}$, we can then use any CP methods for time series, e.g., weighted split CP, to generate weighted nonconformity scores in the form:

$$s_{t+1} = \sum_{i=1}^{t} \tilde{w}_i \cdot \delta_{R_i} + \tilde{w}_{t+1} \cdot \delta_{\infty}, \tag{7}$$

where $R_i = |Y_i - \hat{\mu}(X_i)|$ with $Y_i$ denoting the true value for the $i$-th observation, $\hat{\mu}(X_i)$ representing the predicted value for the $i$-th observation given the base model $\hat{\mu}$. $\tilde{w}_i$ indicates the pre-defined normalized weights, and $\delta_a$ is a point mass function at $a$. The final prediction interval is constructed by calculating a quantile of the weighted nonconformity scores. A detailed illustration of this algorithm, referred to as CP-MDA-Reweighting, is provided in Appendix D.

This naive two-stage approach has several limitations. First, standard weighted split CP relies on predefined weights, making it less adaptable to the diverse requirements of time series data. As shown in Figure 1, inappropriate weights in CP-MDA-Reweighting cause significant under-coverage in the MCAR setting. Additionally, as Zaffran et al. (2023) noted, designing a perfectly matching calibration dataset often excludes too many calibration samples, reducing effectiveness. We further theoretically demonstrate in Theorem F.3 in Appendix F that when the probability of a missing pattern $P_M(m)$ is low, the prediction interval $\widehat{C}_{n,\alpha}(X, m)$ is more likely to become large. Furthermore, the method becomes computationally expensive as the number of features and missing patterns increases. This highlights the need for more effective and efficient CP methods to handle these challenges.

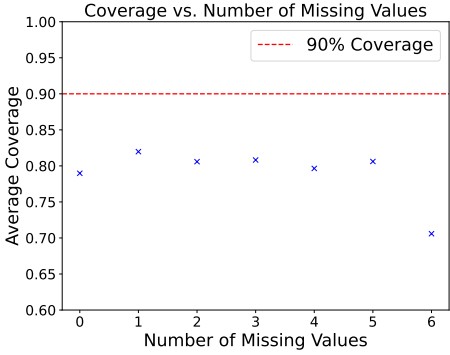

Figure 1: CP-MDA-Reweighting applied to Oil Temperature Data (Zhou et al., 2021) with a miscoverage rate of 0.1.

## 4.2 Our Method: ACP-TSM

The naive approach faces challenges with exponential group growth and fails to address distribution shifts caused by missingness and temporal dependencies. To address these challenges, we utilize all calibration data and leverage the empirical regularity that adjacent time steps tend to admit similar conformal intervals to learn a neighbor-aware correction of weighted split-CP nonconformity scores. This is accomplished by a recurrent neural network (e.g., LSTM) which effectively captures temporal dependencies and incorporates manually designed group-level features to bolster robustness against varying missing mechanisms. Our LSTM's loss functions are designed to optimize conditional coverage and prediction efficiency. To further mitigate training sensitivity to initialization, we introduce adaptive adjustments to stabilize the correction step.

**Conditional Coverage-aware Boosting** We present a loss function that quantifies the conditional coverage rate of any prediction interval. We define groups $G_m$ based on distinct missing value patterns $m \in \mathcal{M}$, where $\mathcal{M}$ represents the set of all possible missing patterns in the dataset. Following the approach of Xie et al.

(2024), for each group $G_m$, the deviation from the target coverage $1 - \alpha$ is computed as the absolute difference between the observed coverage within the group $G_m$ and the desired level:

$$d(C_n(\cdot); G_m) = \left| |G_m|^{-1} \sum_{i \in G_m} \mathbb{1}(Y_i \in C_n(X_i, M_i)) - (1 - \alpha) \right|, \tag{8}$$

where $C_n(X_i, M_i)$ is the resulting prediction interval for $X_i$. This deviation $d(\cdot)$ measures how much the coverage within the group $G_m$ diverges from the target $1 - \alpha$. To ensure balanced coverage across all groups with different missing patterns, we define the overall empirical maximum deviation across missing patterns as:

$$\ell_M(\mathcal{D}; \mathcal{C}) = \max_{m \in \mathcal{M}} d(C_n(\cdot); G_m), \tag{9}$$

where $\mathcal{D}$ represents the dataset with all possible missing patterns and $\mathcal{C}$ represents the CP method. The objective is to minimize $\ell_M(\mathcal{D}; \mathcal{C})$, ensuring that the group-specific coverage for each missing pattern approaches the target coverage level. However, to apply gradient-based optimization techniques, we need to make this loss function differentiable. The definition of empirical coverage involves an indicator function $\mathbb{1}(Y_t \in C_n(X_t))$, which is non-differentiable and unsuitable for optimization. To overcome this, we replace the indicator function with a smooth approximation using the *sigmoid function*, as implemented in Xie et al. (2024). For each group $G_m$, we apply the sigmoid function to the difference between the true values and the adjusted prediction interval bounds:

$$\text{lower\_coverage} = \sigma(c \cdot (Y_t - L_t)); \quad \text{upper\_coverage} = \sigma(c \cdot (U_t - Y_t)), \tag{10}$$

where $L_t$ and $U_t$ are the adjusted prediction interval bounds for $t$, $c$ is a smoothing factor that controls how sharply the sigmoid function behaves. A larger $c$ makes the sigmoid function approximate the hard indicator more closely, while a smaller $c$ makes the transition smoother. This allows us to compute a *smooth empirical coverage* for each group:

$$\text{smooth\_coverage}(G_m) = \frac{1}{|G_m|} \sum_{t \in G_m} \sigma(c \cdot (Y_t - L_t)) \cdot \sigma(c \cdot (U_t - Y_t)), \tag{11}$$

Eq. 11 makes the empirical coverage differentiable, enabling the use of gradient descent for optimization. Next, we compute the deviation between the smooth empirical coverage and the target coverage $1 - \alpha$ for each group:

$$d(G_m) = |\text{smooth\_coverage}(G_m) - (1 - \alpha)| . \tag{12}$$

We track the worst-performing group by calculating the maximum deviation across all groups:

$$\mathcal{L}_{\text{cov}} = \max_{m \in \mathcal{M}} d(G_m). \tag{13}$$

The final loss function is designed to minimize this maximum deviation, ensuring that the worst-performing group's coverage approaches the target coverage level.

**Efficiency-aware Boosting** For improving efficiency, our approach leverages the temporal dependencies present in time series data, i.e., adjacent time points exhibit strong correlations, as consecutive data points often share residual patterns due to their temporal proximity. We empirically confirm this behavior in Figure 2, where we compare the differences in prediction interval lengths across successive time points using vanilla CQR. The results demonstrate that prediction inefficiency has a temporal root: closer time points tend to have consistent residual scales, indicating the presence of a significant temporal structure that can be leveraged to enhance score correction and improve overall prediction efficiency.

Let $\widehat{L}_t$ and $\widehat{U}_t$ represent the adjusted lower and upper bounds for the prediction interval at time $t$:

$$\widehat{L}_t = \mu_{\alpha/2,t} - L_t - q_{\alpha/2}, \quad \widehat{U}_t = \mu_{1-\alpha/2,t} + U_t + q_{1-\alpha/2}, \tag{14}$$

where $\mu_{\alpha/2,t}$ and $\mu_{1-\alpha/2,t}$ are the base quantiles from the Weight Split CP, and $q_{\alpha/2}$ and $q_{1-\alpha/2}$ represent the conformal quantile adjustments. To encourage prediction efficiency using findings in Figure 2, we minimize

the average width of the prediction intervals across all time points:

$$\mathcal{L}_{\text{eff}} = \frac{1}{T} \sum_{t=1}^{T} (\widehat{U}_t - \widehat{L}_t). \tag{15}$$

To avoid generating meaningless intervals (i.e. upper bound is smaller than the lower bound), we impose a regularization term to enforce temporal consistency in the prediction intervals. Since adjacent time points in a time series often exhibit similar patterns, the regularization term penalizes large deviations in the adjusted prediction bounds from the base quantiles $\mu_{\alpha/2,t}$ and $\mu_{1-\alpha/2,t}$. The consistency regularization term $\mathcal{L}_{\text{reg}}$ is defined as:

$$\mathcal{L}_{\text{reg}} = \frac{1}{n} \sum_{t=1}^{n} \left( (\widehat{U}_t - \mu_{1-\alpha/2,t})^2 + (\widehat{L}_t - \mu_{\alpha/2,t})^2 \right). \tag{16}$$

This regularization term ensures that the adjusted bounds remain close to their respective base quantiles, thereby enforcing consistency across consecutive time points. The final loss function $\mathcal{L}_{\text{total}}$ combines the inefficiency loss and the consistency regularization term with a hyperparameter $\lambda_{\text{reg}}$ controlling the trade-off, together with the group conditional coverage loss:

$$\mathcal{L}_{\text{total}} = \mathcal{L}_{\text{cov}} + \mathcal{L}_{\text{reg}} + \lambda_{\text{reg}} \cdot \mathcal{L}_{\text{eff}}. \tag{17}$$

**Post-training LSTM with Dual Loss**
To solve Eqs. 13 and 17, we train an LSTM using a withheld small fraction $\gamma$ of the calibration dataset for the correction procedure, leveraging its ability to capture temporal dependencies and complex patterns in sequential data. The remaining data is used as the standard calibration data for constructing the prediction intervals. The withheld portion is further split into correction calibration and testing sets, simulating the downstream CP step. As a result, we can utilize all calibration data for both the correction and downstream tasks, maximizing the information available. The features fed into LSTM include the original data features, the point predictions from the base model, and the prediction intervals generated using the withheld calibration data.

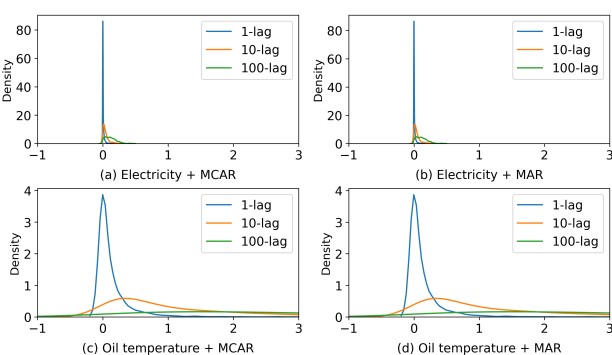

Figure 2: Density plots of the inefficiency gap between timestamp pairs for various lag values across four scenarios: (1) electricity data with MCAR (top-left), (2) electricity data with MAR (top-right), (3) oil temperature data with MCAR (bottom-left), and (4) oil temperature data with MAR (bottom-right).

Inspired by Lipton et al. (2016), we introduce hand-engineered features to handle missing covariates using a binary indicator that signals whether a covariate is missing. Since this indicator remains consistent across instances within a group, we expect LSTM to learn the missing data patterns and adjust prediction intervals accordingly. The features include: (1) a binary indicator for missing values, (2) the mean and standard deviation of the indicator sequence, and (3) the number of jumps in the sequence. These features help the LSTM capture various missing data mechanisms, improving its ability to handle missing values and achieve valid group conditional coverage. We train the LSTM with a dual loss function, as defined in Eq. 13 and 17, to minimize both conditional coverage gap and inefficiency losses, using boosted non-conformity scores as the target output.

**Adaptive Fine-tuning** Given that the training process of NN is sensitive to initialization, the final step of our framework incorporates an adaptive fine-tuning module. We draw on the approach from Gibbs & Candes (2021), which dynamically adjusts $\alpha$ when generating prediction intervals during test time. This adaptation enhances the robustness of the method and reduces the reliance on NN initialization. The update rule for $\alpha$ is given by $\alpha_{t+1} = \alpha_t + \eta \mathbb{1}_{Y_t \in [\hat{L}_t, \hat{U}_t]}$, where $\eta$ is a tuning parameter, and the indicator function $\mathbb{1}_{Y_t \in [\hat{L}_t, \hat{U}_t]}$

adjusts $\alpha$ based on whether the true value $Y_t$ is contained within the prediction interval. We further employ SF-OGD from Bhatnagar et al. (2023) to get rid of hyperparameter tuning. Details of SF-OGD are provided in Appendix E. We summarize ACP-TSM in Algorithm 1, with further details in Appendix A and an overview in Figure 3.

**Theoretical Analysis** Here, we provide a theoretical analysis for our method, demonstrating that coverage will converge with a sufficient number of samples.

**Theorem 4.1.** *Suppose the true radii with coverage guarantee for all test points are bounded: $S_t \in [0, D]$ for all $t \in [T]$. Then, for Algorithm 1 with any initialization $\hat{s}_1 \in [0, D]$, the coverage error at time $T$, denoted by $|\text{CovErr}(T)|$, is bounded as follows:*

$$|\text{CovErr}(T)| \leq \mathcal{O}\left(\alpha^{-2} T^{-1/4} \log T\right),$$

*where $\text{CovErr}(T)$ is defined as:*

$$\text{CovErr}(T) = \left| \frac{1}{T} \sum_{t=1}^{T} \text{err}_t - \alpha \right|.$$

*Here, $\text{CovErr}(T)$ measures the absolute deviation of the average error from the target coverage level $\alpha$ over $T$ time steps. The term $\text{err}_t$ means if the true value falls outside the interval at time $t$, and $\alpha$ refers to the desired miscoverage level.*

$\mathcal{O}(\cdot)$ indicates an asymptotic upper bound on the coverage error, ensuring that the coverage error is controlled as $T$ grows. Detailed proof is in Appendix F. Here, Theorem 4.1 provides an *asymptotic marginal* coverage guarantee. Specifically, under the boundedness assumption that the (unknown) oracle radii that would achieve the target miscoverage level are uniformly bounded, i.e., $S_t \in [0, D]$ for all $t \in [T]$, Algorithm 1 ensures that the *time-average* miscoverage $\frac{1}{T} \sum_{t=1}^{T} \text{err}_t$ converges to the target level $\alpha$. Equivalently, the empirical marginal coverage $1 - \frac{1}{T} \sum_{t=1}^{T} \text{err}_t$ approaches $1 - \alpha$ as $T \to \infty$. This result is closely related to the *adaptive conformal inference* literature (Gibbs & Candes, 2021; Zaffran et al., 2022), which establishes asymptotic coverage guarantees as the horizon $T \to \infty$.

---

**Algorithm 1** ACP-TSM: Conformal Prediction for Time Series with Missing Values

---

**Require:** Training data $\{(X^{(t)}, Y^{(t)}, M^{(t)})\}_{t=1}^{T}$, Calibration data $\{(X^{(t)}, Y^{(t)}, M^{(t)})\}_{t=T+1}^{2T}$, Test point $(X^{\text{test}}, M^{\text{test}})$, Imputation method $\Phi$, Quantile regressor $QR$, Significance $\alpha$, LSTM parameters $\lambda_{\text{reg}}, \eta$
**Ensure:** Prediction interval $\hat{C}_\alpha(X^{\text{test}}, M^{\text{test}})$
1: Fit $\Phi$ on training data and impute missing values.
2: Train $QR$ to estimate $\hat{q}_{\alpha/2}(\cdot)$ and $\hat{q}_{1-\alpha/2}(\cdot)$ on imputed training data.
3: **for** each calibration point $t = T + 1$ to $2T$ **do**
4:     Find past calibration points with subset-matching pattern: Cal(test)
5:     Compute non-conformity scores $s^{(k)}$ using quantile predictions.
6:     Estimate weighted quantile $\hat{Q}_{1-\alpha}$ from $\{s^{(k)}\}_{k \in \text{Cal(test)}}$
7:     Construct interval $(L_t, U_t)$ for each $t$.
8: Train LSTM on calibration intervals to refine $(L_t, U_t)$ using group features.
9: **for** each test point **do**
10:     Apply quantile model to get preliminary interval.
11:     Refine interval using LSTM.
12:     **if** true label $Y_t$ is observed **then**
13:         Compute loss $\ell^{(t)}$ and update $\hat{s}_t$ via online gradient.
    **return** $C_\alpha(X^{\text{test}}, M^{\text{test}})$

---

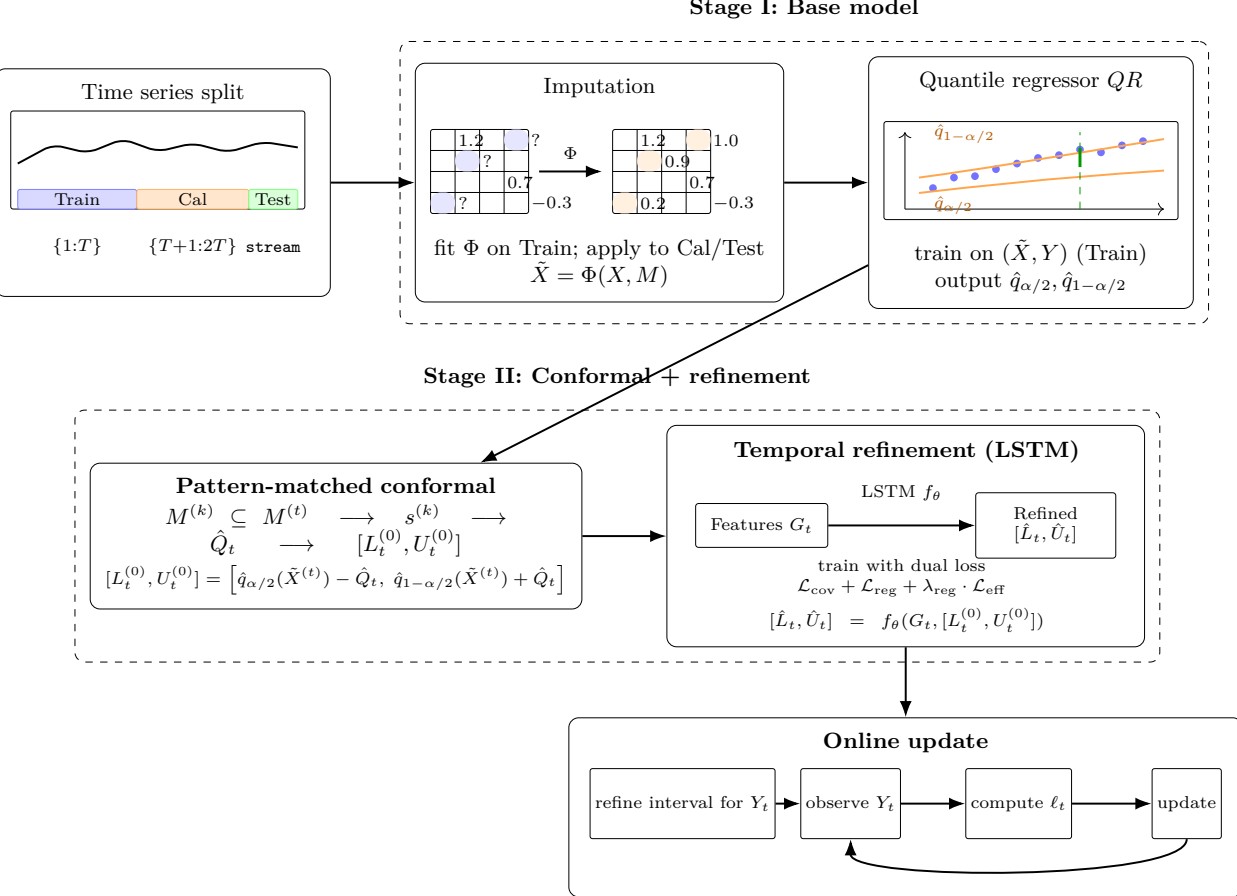

Figure 3: ACP-TSM pipeline: imputation + quantile regression, pattern-matched conformal calibration, and temporal refinement with optional online updates.

## 5 Experiments

### 5.1 Experimental Setup

Following Zaffran et al. (2023), in all experiments, the missing data is first imputed using iterative regression (Pedregosa et al., 2011). The predictive models are then trained on the imputed data concatenated with the mask. The quantile regression algorithm for CQR is a NN for time series data (e.g., LSTM), fitted to minimize the pinball loss in CP. We will consider three missingness settings, MCAR, MAR, and MNAR. Given MNAR constains different missing cases, we will discuss its results separately.

**Datasets.** Our experiments encompass evaluations on three real-world benchmark datasets, allowing for assessment under controlled and natural conditions. They are electricity (Harries et al., 1999) , oil temperature (Zhou et al., 2021), energy appliances (Candanedo, 2017) and air-quality data (De Vito et al., 2008). Details of these datasets can be found in Appendix G. For the first three datasets, we artificially introduce 40% missing values into the training and calibration sets. In contrast, the Air Quality dataset contains naturally occurring missing values and therefore does not require synthetic missingness. All experiments are repeated 5 times with different splits.

**Compared Approaches.** The first set of baselines are methods for CP with missing values and their combination with weighted CP to address the covariate shift: (1) **CQR-MDA-Exact** (Zaffran et al., 2023), a standard CP approach to handle missing values. (2) **CQR-MDA-Reweighting**, the naive approach introduced in Sec. 4.1, a combination of CQR-MDA-Exact with a popular reweighting scheme NexCP (Barber

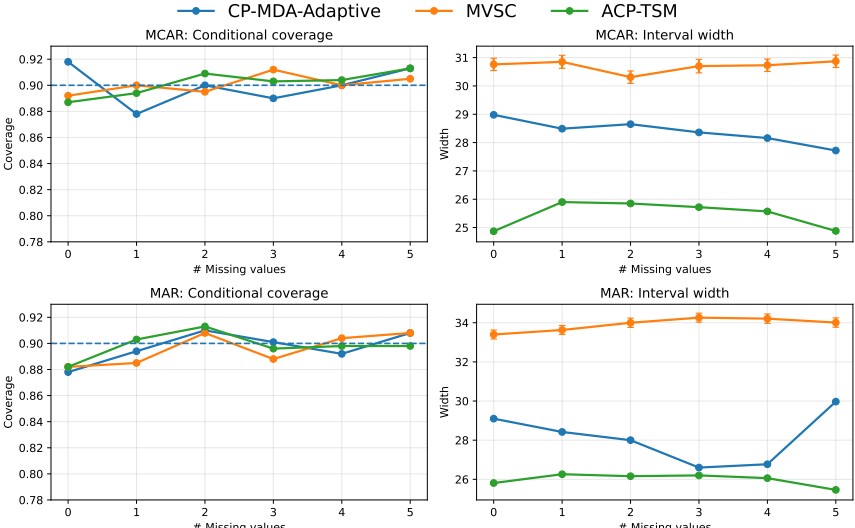

Figure 4: Conditional coverage and interval width by missingness level ($\alpha = 0.1$). Top row: MCAR (coverage, width). Bottom row: MAR (coverage, width). Groups are defined by the number of missing values (0–5). The dashed line indicates the target coverage 0.9.

et al., 2023). (3) **CQR-MDA-Adaptive**, the naive approach introduced in Sec. 4.1, a combination of CQR-MDA-Exact with Adaptive CP (Gibbs & Candes, 2021) for time series. We also compare our approach for CP methods aimed at achieving group conditional coverage: (4) **Multivalid Split Conformal (MVSC)** (Liu & Wu, 2024) adjusts thresholds within each group based on quantiles of scores and updates prediction sets to meet the target coverage condition. To evaluate marginal coverage, we also incorporate results for three popular non-CP-based UQ methods: (5) **Quantile Regression (QR)** (Koenker, 2005), uses a pinball loss to produce quantile scores. It is CQR without the CP adjustment. (6) **Monte Carlo Dropouts (MC dropout)** (Gal & Ghahramani, 2016), turns on dropout during evaluation and produces K predictions. We then take the 90% quantile of the predicted distribution. (7) **Bayesian Neural Networks (Bayesian NN)** (Kendall & Gal, 2017) directly model uncertainty by placing distributions over the network's weights. For QR, MC dropout and Bayesian NN, we use both the training and calibration sets for training. A detailed introduction and implementation details of non-CP methods is provided in Appendix G.1.

**Metrics.** Our analysis includes two common metrics (Angelopoulos & Bates, 2021): (1) *Empirical Coverage Rate (Cov)*, which measures the effectiveness of CP in achieving the theoretically guaranteed coverage, and (2) *Average Prediction Interval Length (Width)* evaluates the efficiency of CP by reflecting the compactness of the prediction intervals. For group conditional coverage, we consider (3) *Average Coverage Gap (CovGap)* (Ding et al., 2024) calculates the mean of the individual group coverage gaps, which quantifies how far group conditional coverage deviates from the desired coverage level, providing a measure of the overall coverage gap across all groups. We also consider (4) the *Winkler score* (Auer et al., 2023), which jointly evaluates coverage and efficiency by penalizing both miscoverage and excessive interval width.

## 5.2 Main Results and Discussions

**Our approach achieves empirical marginal coverage and tight intervals.** We report the marginal coverage, average interval width, and average coverage gap for all methods with a target coverage of 90% in Table 1, focusing on MCAR and MAR scenarios. We observe that first, only our method, CP-MDA-Adaptive, and MVSC consistently achieve the target coverage across all datasets, with our approach showing a 5%-10% improvement in prediction efficiency. This improvement stems from directly minimizing the conditional coverage gap and inefficiency loss. In contrast, CP-MDA-Exact struggles to account for temporal dependencies in time series data, resulting in overly conservative intervals when dealing with many features and missing patterns. Additionally, CP-MDA-Reweighting fails due to improper weight selection. Second, our results

Table 1: Performance of the compared methods for the three benchmark datasets with miscoverage rate $\alpha = 0.1$, under two missing mechanisms. Results were obtained by 5 repetitions with different seeds. Bold numbers indicate the best result, while significant under-coverage is canceled out.

| Methods | | Elec | | OT | | Energy | |
|---|---|---|---|---|---|---|---|
| | | MCAR | MAR | MCAR | MAR | MCAR | MAR |
| CP-MDA-Exact | Cov | $0.885\pm_{.0}$ | $0.883\pm_{.0}$ | $0.665\pm_{.0}$ | $0.666\pm_{.0}$ | $0.952\pm_{.0}$ | $0.955\pm_{.0}$ |
| | Width | $0.602\pm_{.0}$ | $0.593\pm_{.0}$ | $21.30\pm_{.08}$ | $21.13\pm_{.08}$ | $300.56\pm_{5.42}$ | $302.61\pm_{5.42}$ |
| | CovGap | $0.016\pm_{.0}$ | $0.017\pm_{.0}$ | $0.228\pm_{.0}$ | $0.227\pm_{.0}$ | $0.050\pm_{.0}$ | $0.052\pm_{.0}$ |
| | Winkler | $0.703\pm_{.0}$ | $0.705\pm_{.0}$ | $37.22\pm_{.13}$ | $36.93\pm_{.12}$ | $464.78\pm_{8.14}$ | $468.62\pm_{8.02}$ |
| CP-MDA-Reweighting | Cov | $0.899\pm_{.0}$ | $0.902\pm_{.0}$ | $0.807\pm_{.0}$ | $0.807\pm_{.0}$ | $0.898\pm_{.0}$ | $0.898\pm_{.0}$ |
| | Width | $\mathbf{0.584}\pm_{.0}$ | $0.603\pm_{.0}$ | $22.89\pm_{.13}$ | $22.68\pm_{.12}$ | $304.32\pm_{6.52}$ | $304.32\pm_{6.73}$ |
| | CovGap | $\mathbf{0.002}\pm_{.0}$ | $\mathbf{0.004}\pm_{.0}$ | $0.101\pm_{.0}$ | $0.101\pm_{.0}$ | $0.005\pm_{.0}$ | $0.005\pm_{.0}$ |
| | Winkler | $\mathbf{0.692}\pm_{.0}$ | $0.723\pm_{.0}$ | $36.67\pm_{.14}$ | $36.42\pm_{.13}$ | $456.25\pm_{8.13}$ | $458.13\pm_{8.15}$ |
| CP-MDA-Adaptive | Cov | $0.895\pm_{.0}$ | $0.895\pm_{.0}$ | $0.894\pm_{.0}$ | $0.896\pm_{.0}$ | $0.896\pm_{.0}$ | $0.896\pm_{.0}$ |
| | Width | $0.605\pm_{.0}$ | $0.604\pm_{.0}$ | $28.44\pm_{.12}$ | $28.42\pm_{.12}$ | $296.43\pm_{5.52}$ | $296.62\pm_{5.63}$ |
| | CovGap | $0.007\pm_{.0}$ | $0.007\pm_{.0}$ | $0.011\pm_{.0}$ | $0.018\pm_{.0}$ | $0.008\pm_{.0}$ | $0.008\pm_{.0}$ |
| | Winkler | $0.732\pm_{.0}$ | $0.728\pm_{.0}$ | $34.12\pm_{.19}$ | $34.16\pm_{.18}$ | $441.09\pm_{8.13}$ | $441.85\pm_{8.16}$ |
| MVSC | Cov | $0.898\pm_{.0}$ | $0.893\pm_{.0}$ | $0.897\pm_{.0}$ | $0.909\pm_{.0}$ | $0.902\pm_{.0}$ | $0.902\pm_{.0}$ |
| | Width | $0.631\pm_{.0}$ | $0.628\pm_{.0}$ | $30.67\pm_{.18}$ | $34.34\pm_{.18}$ | $576.33\pm_{15.6}$ | $523.74\pm_{15.8}$ |
| | CovGap | $0.003\pm_{.0}$ | $0.010\pm_{.0}$ | $\mathbf{0.002}\pm_{.0}$ | $0.231\pm_{.0}$ | $\mathbf{0.005}\pm_{.0}$ | $0.005\pm_{.0}$ |
| | Winkler | $0.764\pm_{.0}$ | $0.772\pm_{.0}$ | $38.24\pm_{.58}$ | $40.13\pm_{.56}$ | $741.31\pm_{13.4}$ | $698.59\pm_{13.1}$ |
| QR | Cov | $81.28\pm_{.0}$ | $0.757\pm_{.0}$ | $0.868\pm_{.0}$ | $0.870\pm_{.0}$ | $0.922\pm_{.0}$ | $0.922\pm_{.0}$ |
| | Width | $0.445\pm_{.0}$ | $0.493\pm_{.0}$ | $29.56\pm_{2.89}$ | $30.00\pm_{1.88}$ | $314.91\pm_{17.60}$ | $311.01\pm_{10.24}$ |
| | CovGap | $0.152\pm_{.0}$ | $0.143\pm_{.0}$ | $0.030\pm_{.0}$ | $0.029\pm_{.0}$ | $0.021\pm_{.0}$ | $0.022\pm_{.0}$ |
| | Winkler | $0.759\pm_{.0}$ | $0.762\pm_{.0}$ | $33.75\pm_{.84}$ | $33.82\pm_{.82}$ | $446.52\pm_{7.2}$ | $452.36\pm_{7.2}$ |
| MC Dropout | Cov | $0.484\pm_{.01}$ | $0.515\pm_{.0}$ | $0.731\pm_{.0}$ | $0.651\pm_{.0}$ | $0.274\pm_{.01}$ | $0.269\pm_{.01}$ |
| | Width | $0.264\pm_{.0}$ | $0.288\pm_{.0}$ | $15.35\pm_{.44}$ | $15.16\pm_{.35}$ | $58.91\pm_{.89}$ | $59.46\pm_{3.86}$ |
| | CovGap | $0.392\pm_{.0}$ | $0.385\pm_{.0}$ | $0.212\pm_{.0}$ | $0.249\pm_{.0}$ | $0.63\pm_{.0}$ | $0.63\pm_{.0}$ |
| | Winkler | $0.898\pm_{.0}$ | $0.892\pm_{.0}$ | $30.17\pm_{.46}$ | $30.26\pm_{.52}$ | $763.61\pm_{9.8}$ | $753.89\pm_{10.2}$ |
| Bayesian NN | Cov | $0.613\pm_{.0}$ | $0.614\pm_{.0}$ | $0.412\pm_{.0}$ | $0.313\pm_{.0}$ | $0.601\pm_{.0}$ | $0.592\pm_{.0}$ |
| | Width | $0.342\pm_{.0}$ | $0.342\pm_{.0}$ | $10.20\pm_{.0}$ | $8.81\pm_{.0}$ | $90.22\pm_{.07}$ | $90.48\pm_{.03}$ |
| | CovGap | $0.286\pm_{.0}$ | $0.286\pm_{.0}$ | $0.493\pm_{.0}$ | $0.587\pm_{.0}$ | $0.304\pm_{.0}$ | $0.308\pm_{.0}$ |
| | Winkler | $0.823\pm_{.0}$ | $0.825\pm_{.0}$ | $107.05\pm_{2.86}$ | $105.23\pm_{2.45}$ | $636.92\pm_{12.7}$ | $625.92\pm_{13.1}$ |
| ACP-TSM | Cov | $0.899\pm_{.0}$ | $0.900\pm_{.0}$ | $0.902\pm_{.0}$ | $0.890\pm_{.0}$ | $0.894\pm_{.0}$ | $0.896\pm_{.0}$ |
| | Width | $0.592\pm_{.0}$ | $\mathbf{0.588}\pm_{.0}$ | $\mathbf{25.71}\pm_{.04}$ | $\mathbf{25.80}\pm_{.04}$ | $\mathbf{292.53}\pm_{4.32}$ | $\mathbf{293.13}\pm_{4.30}$ |
| | CovGap | $\mathbf{0.002}\pm_{.0}$ | $\mathbf{0.004}\pm_{.0}$ | $0.011\pm_{.0}$ | $\mathbf{0.016}\pm_{.0}$ | $0.009\pm_{.0}$ | $\mathbf{0.003}\pm_{.0}$ |
| | Winkler | $0.695\pm_{.0}$ | $\mathbf{0.684}\pm_{.0}$ | $\mathbf{27.48}\pm_{.23}$ | $\mathbf{27.32}\pm_{.22}$ | $\mathbf{406.91}\pm_{7.23}$ | $\mathbf{408.33}\pm_{6.96}$ |

confirm the theoretical guarantee in Theorem 4.1, demonstrating that our method consistently achieves the specified marginal coverage. Results on real missingness setting in Table 2 further strengthen the effectiveness of our method. Finally, to see how the prediction intervals change over time, we visualize them for the three methods that achieve coverage guarantees in Figure 6 for oil temperature data under MCAR. ACP-TSM generates tighter prediction intervals and adapts quickly to changes in data by leveraging boosted non-conformity scores, while CP-MDA-Adaptive responds more slowly and MVSC is overly conservative.

**Our approach achieves empirical group conditional coverage.** Another observation from Table 1 is that our method achieves the smallest coverage gap across all baselines, underscoring its superior performance in enhancing group conditional coverage. For a more in-depth analysis, we further examine the conditional coverage guarantees of methods that achieve empirical marginal coverage in Table 1. As shown in Figure 4, our method consistently attains group conditional coverage across all groups, with prediction intervals that are 10%-20% narrower than those produced by other methods. For the real missingness setting, the results in Table 2 further demonstrate improved conditional coverage recovery together with consistently shorter prediction intervals. The strength of our approach lies in its ability to minimize both the conditional coverage gap and inefficiency loss, effectively balancing the need for tighter intervals while ensuring reliable coverage.

Table 2: Performance on AirQuality. We report coverage and interval width stratified by missing level (1 and 3), and overall metrics across all test points.

|  | CP-MDA-Adaptive | MVSC | ACP-TSM |
|---|---|---|---|
| *#Missingness = 1* |  |  |  |
| Coverage | 0.865 | 0.913 | 0.891 |
| Interval width | 3.120 | 4.292 | **2.053** |
| *#Missingness = 3* |  |  |  |
| Coverage | 0.875 | 0.923 | 0.875 |
| Interval width | 2.826 | 4.533 | **1.980** |
| *Overall* |  |  |  |
| Coverage | 0.865 | 0.913 | 0.890 |
| Interval width | 3.108 | 4.302 | **2.050** |
| Winkler score | 3.707 | 5.170 | **2.692** |
| Coverage gap | 0.030 | 0.018 | **0.017** |

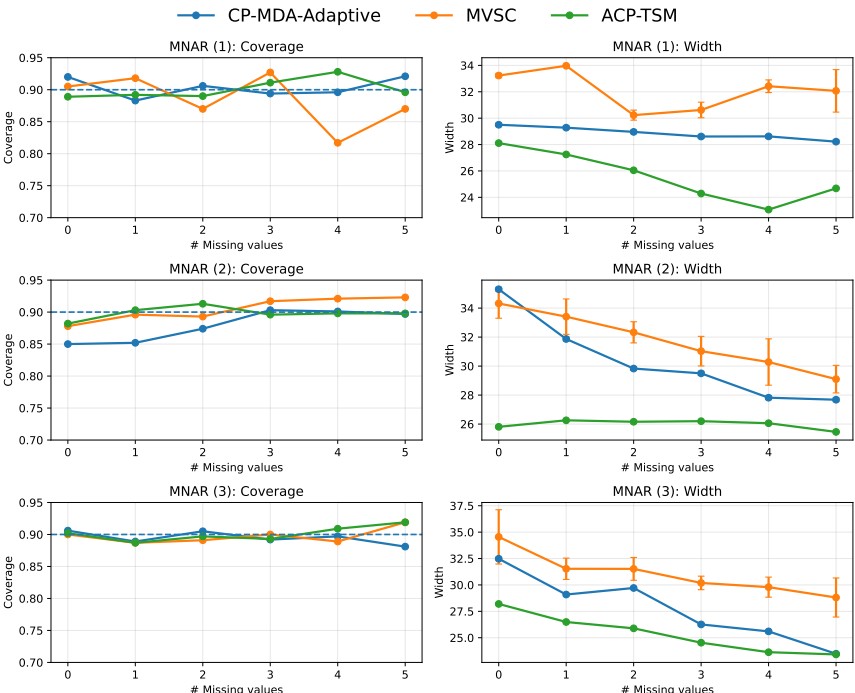

Figure 5: Conditional coverage and interval width by missingness level ($\alpha = 0.1$) for MNAR. Groups are defined by the number of missing values (0–5). The dashed line indicates the target coverage 0.9.

**Our approach is robust to different missing mechanisms.** There can be different types of missingness under MNAR conditions. To better compare the performance under MNAR, we simulate three different settings for the task of predicting oil temperature using fuel load information: (1) missingness related to covariates (e.g., load measurements missing when exceeding 90% capacity), (2) missingness related to the response (e.g., temperature readings missing at extreme levels), and (3) missingness related to time (e.g., data missing during nighttime hours). As shown in Figure 5, our approach consistently maintains group conditional coverage with narrower prediction intervals across all scenarios. While MVSC is considered robust, it remains prone to under-coverage in certain cases, particularly in the first missingness scenario when there are four missing values. This may be due to the complex data structure and the limited amount of data

Table 3: Ablation studies.

| Methods | Ours | w.o. Coverage loss | w.o. Inefficiency Loss | w.o. Adaptive | w.o. Group Features |
|---------|------|--------------------|-----------------------|---------------|---------------------|
| Cov | $89.66\pm_{.0}$ | $86.82\pm_{.0}$ | $92.72\pm_{.0}$ | $88.50\pm_{.0}$ | $89.23\pm_{.0}$ |
| Width | $25.40\pm_{.23}$ | $25.20\pm_{.23}$ | $26.62\pm_{.25}$ | $25.32\pm_{.23}$ | $25.38\pm_{.23}$ |
| CovGap | $0.032\pm_{.0}$ | $0.0564\pm_{.0}$ | $0.0131\pm_{.0}$ | $0.0142\pm_{.0}$ | $0.083\pm_{.0}$ |

Table 4: Step-by-step ablation on the Oil Temperature dataset under MCAR and MAR. We report marginal coverage (%), average coverage gap (%), and average interval width. Each row adds one component on top of the previous variant.

| Setting | Variant | Marginal coverage | Avg. coverage gap | Interval width |
|---------|---------|-------------------|-------------------|----------------|
| | B0: Base | 0.812 | 0.101 | 21.78 |
| | B1: + coverage loss | 0.870 | 0.035 | 25.71 |
| *MCAR* | B2: + inefficiency loss | 0.859 | 0.043 | 25.26 |
| | B3: + adaptive $\alpha$ adjustment | 0.892 | 0.009 | 25.68 |
| | B4: + group features (Full) | 0.896 | 0.001 | 25.70 |
| | B0: Base | 0.809 | 0.101 | 22.72 |
| | B1: + coverage loss | 0.861 | 0.041 | 25.39 |
| *MAR* | B2: + inefficiency loss | 0.862 | 0.042 | 25.38 |
| | B3: + adaptive $\alpha$ adjustment | 0.908 | 0.010 | 25.84 |
| | B4: + group features (Full) | 0.901 | 0.008 | 25.79 |

available for each group. In contrast, our method reliably maintains group conditional coverage, highlighting its robustness in handling all three missingness scenarios.

**Abalation Study.** Finally, we conduct an ablation study in Table 3 to evaluate the effectiveness of the components in ACP-TSM. First, when we replace the coverage loss in Eq.13 with a standard loss , the results show significant under-coverage and a large coverage gap. Secondly, replacing the inefficiency loss in Eq.17 with a standard prediction loss resulted in overly conservative predictions, highlighting the importance of directly modeling inefficiency loss in the correction step. Lastly, when removing components to correct the miscoverage rate or account for additional features related to missing patterns, our approach shows slight under-coverage. We also conduct a step-by-step ablation study of the key components on the Oil Temperature dataset under both MCAR and MAR missingness in Table 4. Adding the coverage loss consistently moves empirical coverage closer to the target level, while adding the inefficiency loss improves efficiency by reducing interval width. Although there are minor fluctuations across variants, the adaptive fine-tuning module and the incorporation of group features further stabilize performance and yield the most reliable overall results across both missingness settings. Additional ablation studies to validate the necessity of combining all components are provided in Appendix G.10.

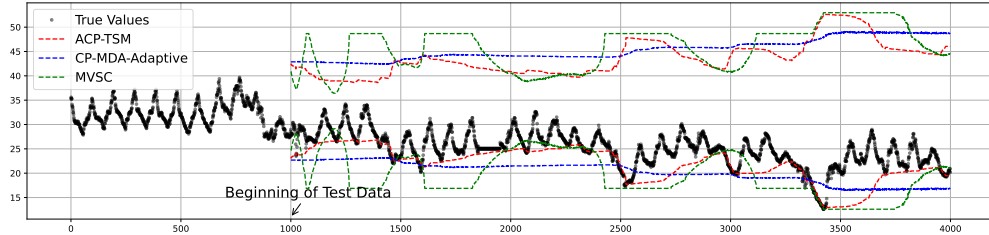

Figure 6: Visualization of Prediction Intervals (i.e., Lower and Upper Bounds) for Oil Temperature Data under MCAR.

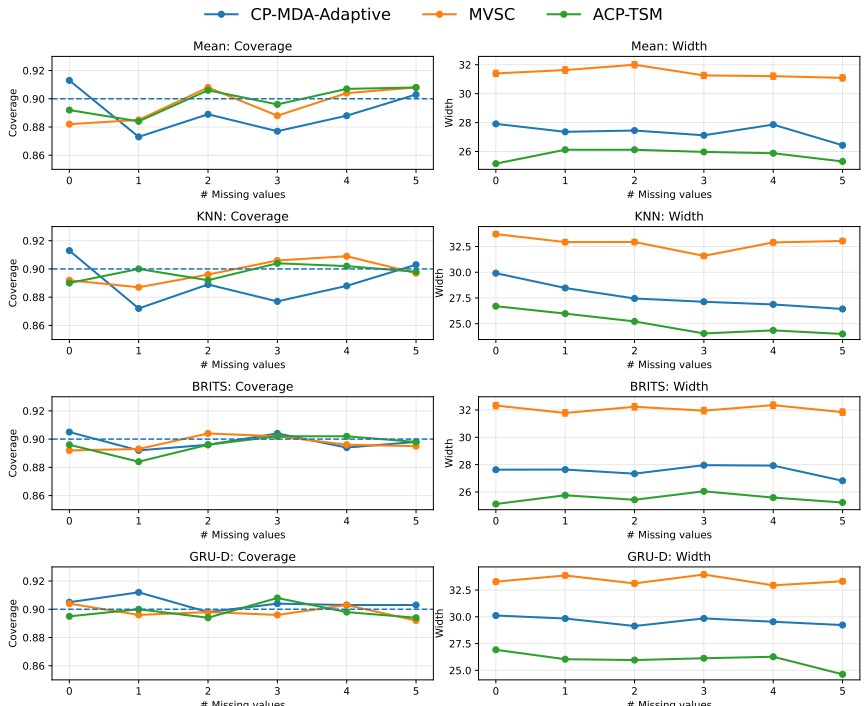

Figure 7: Group-conditional coverage of the compared CP algorithms ($\alpha = 0.1$). Each row corresponds to an imputation method.

**Computational Cost.** Our approach is more computationally efficient than other baseline methods that can achieve empirical coverage, such as CP-MDA-Adaptive and MVSC. For the Oil Temperature dataset, our method finishes in approximately 5 minutes, whereas the baselines take over 20 minutes; on the Energy dataset, our method takes about 40 minutes, while the baselines require 10+ hours (Table 21). The primary computational cost arises from training the NN. We keep costs manageable by using a simple two-layer LSTM (see architecture in Appendix G.11). In contrast, CP-MDA-Adaptive requires sampling a matched calibration dataset for each test point, while MVSC involves adjusting the miscoverage rate for each subgroup, leading to increased resource consumption. Avoiding the sampling process and using the entire calibration dataset can reduce the processing time for CP-MDA-Adaptive to about 3 minutes. More details can be found in Appendix G.

### 5.3  Sensitivity Analysis

In this section, we evaluate ACP-TSM's robustness and sensitivity by varying the imputation algorithm, the degree of missingness, the degree of distribution shift and the hyperparameter settings. In our method, the imputation step is a procedure to map incomplete data to complete data, which can be regarded as part of the training process, as both the imputation algorithm and the base model are fitted only for training data. CP is inherently model-agnostic as a post-hoc uncertainty quantification technique. While different imputation strategies may influence the predictive accuracy of the base model, the validity and informativeness of the ACP-TSM remain unaffected. To validate our claim, we conducted additional experiments using different imputation methods in Figure 7. Specifically, we examined: (1) mean imputation for oil temperature data under MCAR, and (2) KNN imputer for oil temperature data under the MAR. (3) Bidirectional Recurrent Imputation for Time Series (BRITS) (Cao et al., 2018) for oil temperature data under MCAR, and (4) GRU-D (Gated Recurrent Units with Decay) (Che et al., 2018) for oil temperature data under the MAR. Results in Figure 7 demonstrate that our method is robust to imputation methods, consistently providing efficient prediction intervals while maintaining group-conditional coverage.

For different degree of missingness, we extend our experimental results to include scenarios with 0% and 20% missingness. As shown in Tables 5 and 6, our method consistently yields the narrowest prediction intervals while maintaining valid coverage guarantees—even in the absence of missing data—highlighting its practicality for real-world applications.

Table 5: Performance of the compared methods for the three benchmark datasets with miscoverage rate $\alpha =0.1$, under two missing mechanisms with 20% missingness. Results were obtained by 5 repetitions with different seeds. Bold numbers indicate the best result, while significant under-coverage is canceled out.

| Methods | | Elec | | OT Dataset | | Energy | |
|---|---|---|---|---|---|---|---|
| | | MCAR | MAR | MCAR | MAR | MCAR | MAR |
| CP-MDA-Adaptive | Cov | $0.903\pm_{.0}$ | $0.894\pm_{.0}$ | $0.903\pm_{.0}$ | $0.892\pm_{.0}$ | $0.899\pm_{.0}$ | $0.904\pm_{.0}$ |
| | Width | $0.603\pm_{.0}$ | $0.604\pm_{.0}$ | $27.85\pm_{.12}$ | $28.23\pm_{.12}$ | $298.33\pm_{4.83}$ | $298.73\pm_{4.88}$ |
| | CovGap | $0.007\pm_{.0}$ | $0.007\pm_{.0}$ | $0.012\pm_{.0}$ | $0.016\pm_{.0}$ | $0.006\pm_{.0}$ | $0.007\pm_{.0}$ |
| MVSC | Cov | $0.903\pm_{.0}$ | $0.902\pm_{.0}$ | $0.896\pm_{.0}$ | $0.895\pm_{.0}$ | $0.898\pm_{.0}$ | $0.904\pm_{.0}$ |
| | Width | $0.633\pm_{.0}$ | $0.629\pm_{.0}$ | $32.23\pm_{.18}$ | $32.35\pm_{.18}$ | $565.37\pm_{14.7}$ | $545.67\pm_{14.7}$ |
| | CovGap | $0.003\pm_{.0}$ | $0.08\pm_{.0}$ | $\mathbf{0.005}\pm_{.0}$ | $0.212\pm_{.0}$ | $\mathbf{0.007}\pm_{.0}$ | $0.009\pm_{.0}$ |
| ACP-TSM | Cov | $0.902\pm_{.0}$ | $0.898\pm_{.0}$ | $0.902\pm_{.0}$ | $0.897\pm_{.0}$ | $0.903\pm_{.0}$ | $0.898\pm_{.0}$ |
| | Width | $\mathbf{0.598}\pm_{.0}$ | $\mathbf{0.592}\pm_{.0}$ | $\mathbf{26.42}\pm_{.04}$ | $\mathbf{26.35}\pm_{.04}$ | $\mathbf{293.58}\pm_{4.25}$ | $\mathbf{293.46}\pm_{4.22}$ |
| | CovGap | $\mathbf{0.002}\pm_{.0}$ | $\mathbf{0.006}\pm_{.0}$ | $0.008\pm_{.0}$ | $\mathbf{0.012}\pm_{.0}$ | $0.009\pm_{.0}$ | $\mathbf{0.006}\pm_{.0}$ |

Table 6: Performance of the compared methods for the three benchmark datasets with miscoverage rate $\alpha = 0.1$, under two missing mechanisms. Results were obtained by 5 repetitions with different seeds. Bold numbers indicate the best result, while significant under-coverage is canceled out.

| Methods | | Elec | | OT Dataset | | Energy | |
|---|---|---|---|---|---|---|---|
| | | MCAR | MAR | MCAR | MAR | MCAR | MAR |
| CP-MDA-Adaptive | Cov | $0.892\pm_{.0}$ | $0.901\pm_{.0}$ | $0.897\pm_{.0}$ | $0.909\pm_{.0}$ | $0.904\pm_{.0}$ | $0.906\pm_{.0}$ |
| | Width | $0.612\pm_{.01}$ | $0.618\pm_{.01}$ | $28.50\pm_{.32}$ | $28.59\pm_{.31}$ | $303.03\pm_{4.03}$ | $307.49\pm_{4.07}$ |
| | CovGap | $0.015\pm_{.0}$ | $0.012\pm_{.0}$ | $0.014\pm_{.0}$ | $\mathbf{0.005}\pm_{.0}$ | $0.012\pm_{.0}$ | $0.012\pm_{.0}$ |
| MVSC | Cov | $0.905\pm_{.0}$ | $0.902\pm_{.0}$ | $0.900\pm_{.0}$ | $0.897\pm_{.0}$ | $0.908\pm_{.0}$ | $0.910\pm_{.0}$ |
| | Width | $0.623\pm_{.00}$ | $0.627\pm_{.0}$ | $32.78\pm_{.31}$ | $32.26\pm_{.29}$ | $509.69\pm_{4.10}$ | $506.80\pm_{4.07}$ |
| | CovGap | $0.010\pm_{.0}$ | $0.008\pm_{.0}$ | $0.011\pm_{.0}$ | $0.009\pm_{.0}$ | $\mathbf{0.007}\pm_{.0}$ | $0.008\pm_{.0}$ |
| ACP-TSM | Cov | $0.899\pm_{.0}$ | $0.903\pm_{.0}$ | $0.905\pm_{.0}$ | $0.899\pm_{.0}$ | $0.900\pm_{.0}$ | $0.901\pm_{.0}$ |
| | Width | $\mathbf{0.605}\pm_{.01}$ | $\mathbf{0.603}\pm_{.01}$ | $\mathbf{26.80}\pm_{.30}$ | $\mathbf{26.92}\pm_{.30}$ | $\mathbf{297.68}\pm_{3.98}$ | $\mathbf{299.23}\pm_{3.99}$ |
| | CovGap | $\mathbf{0.008}\pm_{.0}$ | $\mathbf{0.007}\pm_{.0}$ | $\mathbf{0.006}\pm_{.0}$ | $\mathbf{0.005}\pm_{.0}$ | $0.009\pm_{.0}$ | $\mathbf{0.007}\pm_{.0}$ |

To further assess robustness under nonstationarity and potential regime changes, we conduct two complementary evaluations on Oil Temperature data. First, we report performance separately on early versus late time periods. As shown in Table 7, interval performance is relatively stable in earlier segments but degrades noticeably in the later segment, consistent with the distributional shift in Figure 6. We attribute this degradation to a violation of exchangeability between the calibration and test points: when test points drift away from the calibration distribution, even conservative intervals may exhibit under-coverage. Overall, these results highlight the practical importance of the exchangeability assumption underlying conformal prediction and motivate robustness to temporal distribution shift for real-world deployment.

Second, we stratify test points into regimes according to a simple local change score

$$d_t = |y_{t-1} + y_{t+1} - 2y_t|,$$

and define *smooth*, *intermediate*, and *abrupt* regimes via empirical quantile thresholds of $\{d_t\}$. We then report coverage and average interval width within each regime (Table 8), which directly diagnoses when temporal smoothing helps or fails under local distributional shift. Performance is broadly consistent across regimes: coverage remains close to the target level in both smooth and abrupt segments, while the intermediate

Table 7: Coverage and interval width across three chronological test segments.

| Segment | Coverage | Width |
|---------|----------|-------|
| Seg1 | 0.9110 | 16.7659 |
| Seg2 | 0.8650 | 20.2047 |
| Seg3 | 0.8340 | 28.4535 |

Table 8: Coverage and interval width by shift regime.

| Regime | $n$ | Coverage | Width |
|--------|-----|----------|-------|
| Overall | 3000 | 0.9030 | 18.6554 |
| Smooth | 1027 | 0.9049 | 18.6400 |
| Intermediate | 1008 | 0.8978 | 18.6054 |
| Abrupt | 965 | 0.9065 | 18.7257 |

regime exhibits a slight under-coverage; meanwhile, the average width is nearly unchanged across regimes. This suggests that the proposed temporal refinement is not overly sensitive to local shift intensity in this dataset. Further discussion of practical deployment and maintenance under distribution shift is deferred to Appendix H.

Finally, in our approach, we reserve a portion of the data as a validation set to tune the hyperparameters, following a procedure similar to Huang et al. (2024). The parameter ranges in our experiments are shown in Table 9. The results in Table 10 indicate that our method is robust to these hyperparameter choices, largely due to the fine-tuning component integrated within our framework. Nevertheless, appropriate tuning can further improve performance. In particular, $\lambda_{\mathrm{reg}}$ controls the tradeoff between reducing the conditional coverage gap and improving prediction efficiency: if $\lambda_{\mathrm{reg}}$ is too large, the optimization may over-emphasize efficiency and shrink intervals aggressively, which can slow down (or hinder) the reduction of the coverage gap, whereas if $\lambda_{\mathrm{reg}}$ is too small, the coverage term dominates and leads to conservative (wide) intervals with diminishing returns in efficiency. Specifically, we aim to scale the two loss terms to similar magnitudes to ensure effective training. Intuitively, while the scale of the coverage gap remains relatively consistent across datasets, prediction efficiency can vary significantly. Our results indicate that when the error term is larger, a smaller regularization parameter is required. In cases of severe inefficiency, reducing $\lambda_{reg}$ helps balance the contributions of the conditional coverage loss and the prediction efficiency loss. To support this claim, we include additional experiments on all three datasets under the MCAR setting. Results in Table 10 show that the performance slightly changes as $\lambda_{\mathrm{reg}}$ varies, and the changes primarily stem from the imbalance between the two loss components. As a practical solution, we observe a simple yet effective heuristic: the product of the interval width and $\lambda_{\mathrm{reg}}$ tends to lie within the range of 10 to 100 across datasets. To further validate this observation, Table 11 presents a more detailed hyperparameter search on the Energy dataset. The results confirm that the model performance remains stable and consistent when this product falls within the 10 to 100 range. In addition, the smoothing factor $c$ controls the sharpness of the surrogate used in the coverage-related component: larger $c$ produces a smoother surrogate that improves numerical stability and is closer to the original non-smooth objective. Finally, the post-hoc LSTM is trained after fitting the base predictor and acts as a lightweight calibrator that maps recent prediction/coverage statistics to an adjustment, which improves stability because it avoids jointly optimizing the predictive model and calibration dynamics.

Table 9: Parameter ranges for the LSTM model.

| Param. | Range |
|--------|-------|
| $\lambda_{reg}$ | [0.1,1,10,100] |
| $c$ | [100,1000] |

Table 10: Comparison of coverage rates and prediction interval widths under MCAR across multiple datasets with different hyperparameters.

| Dataset | $c$ | $\lambda_{reg} = 0.1$ | | $\lambda_{reg} = 1$ | | $\lambda_{reg} = 10$ | | $\lambda_{reg} = 100$ | |
|---------|-----|------|-------|------|-------|------|-------|------|-------|
| | | Cov | Width | Cov | Width | Cov | Width | Cov | Width |
| Oil | 100 | 0.898 | 25.68 | 0.902 | 25.71 | 0.897 | 25.76 | 0.902 | 25.83 |
| | 1000 | 0.902 | 25.78 | 0.897 | 25.63 | 0.904 | 25.94 | 0.902 | 25.82 |
| Elec | 100 | 0.901 | 0.602 | 0.899 | 0.608 | 0.898 | 0.601 | 0.899 | 0.592 |
| | 1000 | 0.903 | 0.599 | 0.900 | 0.595 | 0.901 | 0.596 | 0.899 | 0.604 |
| Energy | 100 | 0.894 | 292.5 | 0.894 | 299.8 | 0.868 | 277.6 | 0.852 | 268.2 |
| | 1000 | 0.899 | 292.7 | 0.900 | 302.4 | 0.865 | 276.3 | 0.861 | 270.9 |

Table 11: Comparison of coverage rates and prediction interval widths for Energy Data under MCAR with different hyperparameters

| $\lambda_{reg}$ | 0.05 | | 0.1 | | 0.2 | | 0.5 | |
|-----------------|----------|-------|----------|-------|----------|-------|----------|-------|
| c | Coverage | Width | Coverage | Width | Coverage | Width | Coverage | Width |
| 100 | 0.896 | 293.12 | 0.894 | 292.5 | 0.899 | 291.7 | 0.882 | 288.78 |
| 1000 | 0.901 | 292.58 | 0.899 | 292.7 | 0.902 | 292.2 | 0.898 | 289.31 |

## 6 Conclusions

We propose ACP-TSM, a lightweight post-hoc conformal prediction method for time series with missing values. Starting from any base forecaster, we train a small LSTM on the calibration set to correct nonconformity scores using missingness-aware features and nearby-step context. The LSTM optimizes a dual objective that penalizes group-conditional coverage gaps while minimizing interval width (inefficiency), yielding tighter sets at the target coverage. The procedure is model-agnostic and imputation-agnostic, incurs minimal overhead (no retraining of the base model), and preserves standard split-CP marginal coverage by construction; we further provide conditions under which group-conditional coverage improves. Empirically, ACP-TSM achieves the desired coverage with competitive prediction efficiency across a range of missingness levels and imputers.

Our procedure requires a held-out split for calibration and hyperparameter selection. When datasets are relatively small, reserving a calibration set reduces the effective sample size for model fitting. In very small-sample regimes, some prior methods can be less sensitive to this issue by using simpler calibration choices, while having comparable computational cost. In addition, our experiments primarily use semi-synthetic missingness, which may not fully reflect all real-world missingness pathologies. Finally, in this work we focus on time series with missing covariates; extending ACP-TSM to other forms of non-exchangeability (change point detection) and data imperfections (e.g., label noise or adversarial perturbations) is an important direction for future work.

### Acknowledgments

This work is supported by the National Science Foundation (NSF) Grant #2312862, NSF-Simons SkAI Institute, NSF CAREER #2440542, NSF #2533996, National Institutes of Health (NIH) #R01AG091762, NSF ACCESS Computing Resources, NAIRR, a Google Research Scholar Award, and Cisco gift grant.

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

# Appendix

## A   ACP-TSM

The ACP-TSM algorithm generates reliable prediction intervals for time series with missing values through four steps:

1. From Lines 1-3, train an imputation model $\Phi$ to fill missing entries in the training, calibration, and test data.

2. From Lines 4-11, use CP-MDA-Reweighting to calculate weighted non-conformity scores and generate corresponding prediction intervals for the calibration data.

3. From Lines 12-15, refine these intervals further by training an LSTM model with dual loss functions.

4. From Lines 16-23, dynamically update prediction intervals for test data using observed outcomes and adaptive fine-tuning.

For simplicity, we focus on one-step forecasting, where one single value is predicted at each time step using all prior data. However, our algorithm can be readily extended to handle multi-step time series forecasting.

## B   Missing Mechanisms

In this section we formally introduce three well-known missing data mechanisms proposed by Rubin (1976) .

**Definition B.1.** Missing Completely at Random (MCAR): Missingness is independent of both observed and unobserved values of the data. Formally,

$$\mathbb{P}(M = m|X) = \mathbb{P}(M = m).$$

While MCAR simplifies the analysis, it is often unrealistic in time series data where missingness tends to exhibit structure.

**Definition B.2.** Missing At Random (MAR): Missingness is dependent only on the observed portion of the data, not on the unobserved values:

$$\mathbb{P}(M = m|X) = \mathbb{P}(M = m|X_{\mathrm{obs}(m)}),$$

where $X_{\mathrm{obs}(m)}$ are the observed components of $X$. MAR is more applicable in many real-world scenarios where missingness is correlated with the observed data.

---

**Algorithm 2** ACP-TSM: Enhanced Conformal Prediction for Time Series with Missing Values

---

**Require:** Training data $\{(X^{(t)}, Y^{(t)}, M^{(t)})\}_{t=1}^{T}$, Calibration data $\{(X^{(t)}, Y^{(t)}, M^{(t)})\}_{t=T+1}^{2T}$, Test data $\{(X^{(test)}, M^{(test)})\}$, Imputation algorithm $\Phi$, Quantile regression method $QR$, Significance level $\alpha \in (0, 1)$, LSTM correction parameters $\{\lambda_{\text{reg}}, \eta\}$, learning rate $eta$ and initialization $\hat{s}_1$

**Ensure:** Prediction interval $\hat{C}_\alpha(X^{(test)}, M^{(test)})$

1: Fit the imputation function $\Phi(\cdot)$ using the training data: $\Phi(\cdot) \leftarrow I\left(\left\{\left(X^{(t)}, M^{(t)}\right)\right\}_{t=1}^{T}\right)$

2: Impute the training data using the fitted $\Phi(\cdot)$: $X_{\text{imp}}^{(t)} \leftarrow \Phi\left(X^{(t)}, M^{(t)}\right)_{t=1}^{T}\}$

3: Fit quantile regression using the training data:

$$\hat{q}_{\alpha/2}(\cdot) \leftarrow QR\left(\left\{\left(X_{\text{imp}}^{(t)}, Y^{(t)}\right)_{t=1}^{T}\right\}, \alpha/2\right) \quad \hat{q}_{1-\alpha/2}(\cdot) \leftarrow QR\left(\left\{\left(X_{\text{imp}}^{(t)}, Y^{(t)}\right)_{t=1}^{T}\right\}, 1-\alpha/2\right)$$

4: Apply the CP-MDA-Reweighting method to generate prediction intervals for Calibration data

5: **for** each data point $\{(X^{(t)}, Y^{(t)}, M^{(t)})\}_{t=T+1}^{2T}$ **do**

6:     Define the augmented calibration set: $\text{Cal(test)} \leftarrow \{k \in \text{Cal} \mid M^{(k)} \subseteq M^{(t)} \text{ and } k < t\}$.

7:     Impute the calibration data using the fitted $\Phi(\cdot)$: $X_{\text{imp}}^{(t)} \leftarrow \Phi\left(X^{(t)}, M^{(t)}\right)_{t=T+1}^{2T}\}$

8:     Non-conformity scores: $s^{(k)} = \max\left(\hat{q}_{\alpha/2}(X_{\text{imp}}^{(k)}) - Y^{(k)}, Y^{(k)} - \hat{q}_{1-\alpha/2}(X_{\text{imp}}^{(k)})\right), \forall k \in \text{Cal(test)}$.

9:     Calculate the weighted quantile: $\hat{Q}_{1-\alpha}(S_{\text{Cal}}) \leftarrow \text{Quantile}\left(\sum_{k \in \text{Cal(test)}} w_k \cdot \delta_{s^{(k)}} + w_{t+1} \cdot \delta_\infty\right)$.

10:     Construct the prediction intervals $(L_t, U_t)$:

$$L_t = \hat{q}_{\alpha/2}(\Phi(X^{(test)}, M^{(test)}) - \hat{Q}_{1-\alpha}(S_{\text{Cal}}) \quad U_t = \hat{q}_{1-\alpha/2}(\Phi(X^{(test)}, M^{(test)}) + \hat{Q}_{1-\alpha}(S_{\text{Cal}}).$$

11: Split the calibration data into two subsets: $\text{Cal(train)} = \gamma \cdot \text{Cal}, \text{Cal(test)} = (1-\gamma) \cdot \text{Cal}$.

12: Calculate features for capturing group information.

13: Train the LSTM on Cal(train) to predict refined intervals $(\hat{L}_t, \hat{U}_t)$ with dual loss functions.

14: Use the remaining calibration data Cal(test) for downstream conformal prediction with the refined intervals $(\hat{L}_t, \hat{U}_t)$.

15: **Adaptive Fine-Tuning:**

16: **for** each test data point $(X^{(test)}, M^{(test)})$ **do**

17:     Use CQR to generate prediction intervals with Cal(test)

18:     Correct the prediction intervals using the trained LSTM

19:     Observe true label $Y_t^{(test)}$ and compute true radius: $S_t = \inf\{s \in \mathbb{R} : Y_t^{(test)} \in \hat{C}_t(X^{(test)}, s)\}$.

20:     Compute loss: $\ell^{(t)}(s) = \ell_{1-\alpha}(\hat{C}_t(X^{(test)}, s))$.

21:     Update predicted radius: $\hat{s}_{t+1} = \hat{s}_t - \eta \dfrac{\nabla \ell^{(t)}(S_t)}{\sqrt{\sum_{i=1}^{t} \|\nabla \ell^{(i)}(S_i)\|_2^2}}$.

**return** $\hat{C}_\alpha(X^{(test)}, M^{(test)}) = [\hat{L}_{test}, \hat{U}_{test}]$

---

**Definition B.3.** Missing Not At Random (MNAR): All scenarios outside MCAR and MAR fall under MNAR. Here, the missingness depends on both the observed and unobserved components of the data. Formally,

$$\mathbb{P}(M = m | X) = \mathbb{P}(M = m | X_{\text{obs}(m)}, X_{\text{mis}(m)}),$$

where $X_{\text{mis}(m)}$ are the missing components of $X$. This is often the most challenging case to handle, as the missingness is directly linked to the unobserved values, which is common in many time series settings.

## C   Impute-then-predict

Handling missing values is a crucial challenge in machine learning algorithms, as most predictive models require complete data. A common approach to address this issue is the impute-then-predict framework (Le Morvan et al., 2021). In this method, an imputation function $\Phi$ is used to map observed values to plausible missing values, ensuring that the model can make valid predictions despite the incomplete data. Formally, for a feature $X_j$ with missingness pattern $M \in \mathcal{M}$, the imputation function $\phi^m$ is defined as: $\phi^m : \mathbb{R}^{|\text{obs}(m)|} \longrightarrow \mathbb{R}^{|\text{mis}(m)|}$, where $\phi^m$ takes as input the observed values $X_{\text{obs}(m)}$ and outputs imputed values $X_{\text{mis}(m)}$. Then, the imputation function $\Phi$ for the entire dataset belongs to the space of functions $\mathcal{F}^I$, where:

$$\mathcal{F}^I := \big\{ \Phi : \mathbb{R}^d \times \mathcal{M} \to \mathbb{R}^d \ : \ \forall j \in [1, d],$$
$$\Phi_j(X, M) = X_j \mathbb{1}_{M_j = 0} + \phi_j^M(X_{\text{obs}(m)}) \mathbb{1}_{M_j = 1} \big\}.$$

This imputation step ensures that the dataset is transformed into a format that can be handled by standard predictive models. In many practical applications, $\Phi$ is derived as the output of an imputation algorithm trained on the available data $\{(X^{(k)}, M^{(k)})\}_{k=1}^n$.

## D   CP-MDA-Reweighting

The pseudocode for CP-MDA-Reweighting is shown in Algorithm 3.

---
**Algorithm 3** CP-MDA-Reweighting
---

1: **Input:** Imputation algorithm $\mathcal{I}$, quantile regression algorithm $\hat{q}$, significance level $\alpha$, training set $\{(X^{(t)}, Y^{(t)})\}_{t=1}^T$, test point $(X^{\text{test}}, M^{\text{test}})$, pre-defined weights $\{w_t\}_{t=1}^{n+1}$

2: **Output:** Prediction interval $\hat{C}_\alpha(X^{\text{test}}, m)$

3: Randomly split $\{1, \ldots, T\}$ into two disjoint sets $Tr$ and $Cal$

4: Fit the imputation function: $\phi(\cdot) \leftarrow \mathcal{I}\big(\{(X^{(t)}, M^{(t)}) \mid t \in Tr\}\big)$

5: Impute the training set: $X_{\text{imp}}^{(t)} \leftarrow \{\phi(X^{(t)}, M^{(t)}) \mid t \in Tr\}$

6: Fit algorithm $\hat{q}$:
$$\hat{q}_{\alpha/2}(\cdot) \leftarrow QR\left(\{(X_{\text{imp}}^{(t)}, Y^{(t)}) \mid t \in Tr\}, \alpha/2\right)$$
$$\hat{q}_{1-\alpha/2}(\cdot) \leftarrow QR\left(\{(X_{\text{imp}}^{(t)}, Y^{(t)}) \mid t \in Tr\}, 1 - \alpha/2\right)$$

7: Generate an augmented calibration set: $\text{Cal}(\text{test}) \leftarrow \{t \in \text{Cal} \text{ such that } m^{(t)} \subset m^{(test)}\}$

8: **for** $t \in \text{Cal}(\text{test})$ **do**

9:     $\tilde{m}^{(t)} \leftarrow m^{(t)}$

10: **for** $t \in \text{Cal}(\text{test})$ **do**

11:     Impute the calibration set: $\{X_{\text{imp}}^{(t)}\}_{t \in \text{Cal}(\text{test})} \leftarrow \{\phi(X^{(t)}, \tilde{M}^{(t)})\}_{t \in \text{Cal}(\text{test})}$

12:     Set $s^{(t)} = \max\left(\hat{q}_{\alpha/2}(x_{\text{imp}}^{(t)}) - y^{(t)}, \ y^{(t)} - \hat{q}_{1-\alpha/2}(x_{\text{imp}}^{(t)})\right)$

13: Calculate the weighted quantile $\hat{Q}_{1-\alpha}(\sum_{t \in \text{Cal}(\text{test})} \tilde{w}_t \cdot \delta_{s^{(t)}} + \tilde{w}_{t+1} \cdot \delta_\infty)$

14: $\hat{Q}_{1-\alpha}(S_{\text{Cal}}) = \hat{C}_\alpha(X^{\text{test}}, M^{\text{test}}) \leftarrow \{y \mid s(y, (X^{\text{test}}, M^{\text{test}})) \leq \hat{Q}_{1-\alpha}(S_{\text{Cal}})\}$

---

# E Scale-Free Online Gradient Descent (SF-OGD)

Scale-Free Online Gradient Descent (SF-OGD) is a general algorithm for online learning. In the setting of ACI, it was originally a sub-algorithm of SAOCP, but it was found to have good performance by itself, so we discuss its properties here. The algorithm updates $\alpha_t$ with a gradient descent step where the learning rate adapts to the scale of the previously observed gradients. The pseudocode for SF-OGD is shown in Algorithm 4.

---

**Algorithm 4** SF-OGD, algorithm 2 in Bhatnagar et al. (2023).

---

1: **Input:** $\alpha \in (0,1)$, learning rate $\eta > 0$, initial $\hat{s}_1 \in \mathbb{R}$
2: **for** $t \geq 1$ **do**
3:     Observe input $X_t \in \mathcal{X}$
4:     Return prediction set $\hat{C}_t(X_t, \hat{s}_t)$
5:     Observe true label $Y_t \in \mathcal{Y}$ and compute true radius $S_t = \inf\{s \in \mathbb{R} : Y_t \in \hat{C}_t(X_t, s)\}$
6:     Compute loss $\ell^{(t)}(\cdot) = \ell_{1-\alpha}(S_t, \cdot)$
7:     Update predicted radius $\hat{s}_{t+1} = \hat{s}_t - \eta \dfrac{\nabla \ell^{(t)}(\hat{s}_t)}{\sqrt{\sum_{i=1}^{t} \|\nabla \ell^{(t)}(\hat{s}_i)\|_2^2}}$

---

# F MISSING PROOFS

## F.1 Proof of Theorem 4.1

**Lemma F.1.** *(Bounded iterates for SF-OGD). Suppose the true radii are bounded: $S_t \in [0, D]$ for all $t \in [T]$. Then Algorithm 3 with any initialization $\hat{s}_1 \in [-\eta, D + \eta]$ and learning rate $\eta > 0$ admits bounded iterates:*

$$\hat{s}_t \in [-\eta, D + \eta] \quad \text{for all } t \in [T].$$

*Proof.* The gradient in algorithm 3 can be evaluated and has the following simple form:

$$\nabla \ell^{(t)}(\hat{s}_t) = \alpha - 1\,[\hat{s}_t < S_t] = \alpha - \mathrm{err}_t \in \{-(1-\alpha), \alpha\} \subset [-1, 1] \text{for all } t \in [T]. \tag{18}$$

Therefore, Algorithm 3 satisfies for any $t \geq 1$ that

$$|\hat{s}_{t+1} - \hat{s}_t| = \eta \left| \frac{\alpha - \mathrm{err}_t}{\sqrt{\sum_{\tau=1}^{t}(\alpha - \mathrm{err}_\tau)^2}} \right| \leq \eta. \tag{19}$$

We prove the lemma by contradiction. Suppose there exists some $t$ such that $\hat{s}_t \notin [-\eta, D + \eta]$. Let $t \geq 2$ be the smallest such time index (abusing notation slightly). Suppose $\hat{s}_t > D + \eta$, then by Equation 19 we must have $\hat{s}_{t-1} > D$ but $\hat{s}_{t-1} \leq D + \eta$. Note that $\hat{s}_{t-1} > D \geq S_{t-1}$ by our precondition, so that the (t-1)-th prediction set must cover and thus $\mathrm{err}_{t-1} = 0$. Therefore by the algorithm update we have

$$\hat{s}_t = \hat{s}_{t-1} - \eta \frac{\alpha - \mathrm{err}_{t-1}}{\sqrt{\sum_{\tau=1}^{t-1}(\alpha - \mathrm{err}_\tau)^2}} < \hat{s}_{t-1} \leq D + \eta, \tag{20}$$

contradicting with our assumption that $\hat{s}_t > D + \eta$. A similar contradiction can be derived for the case where $\hat{s}_t < -\eta$. This proves the desired result. $\qquad\square$

**Lemma F.2.** *Suppose the sequence $\{a_t\}_{t \in [T]} \in \mathbb{R}$ satisfies $\alpha \leq |a_t| \leq 1$ for some $\alpha > 0$, and*

$$\left| \sum_{t=t_0+1}^{t_f} \frac{a_t}{\sqrt{\sum_{s=1}^{t} a_s^2}} \right| \leq M \quad \text{for any} \quad 0 \leq t_0 < t_f \leq T.$$

*Then we have*

$$\left| \frac{1}{T} \sum_{t=1}^{T} a_t \right| \le 2(M + 1 + \alpha^{-2} \log T) T^{-1/4}.$$

*Proof.* The proof builds on a grouping argument. Define integers

$$L = \lceil T^{\beta} \rceil, \quad K = \lceil T/L \rceil \le T^{1-\beta} + 1, \tag{21}$$

where $\beta \in (0, 1)$ is a parameter to be chosen. For any $k \in [K]$, define the $k$-th group to be

$$G_k = \{t_{k-1} + 1, \dots, t_k\} := \{(k-1)L + 1, \dots, \min\{kL, T\}\}, \tag{22}$$

so that we have $\bigcup_{k=1}^{K} G_k = [T]$, $|G_k| = L$ for all $k \in [K-1]$, and $|G_K| \le L$.

Next, for any fixed $k \ge 2$, define sums

$$S_k := \sum_{t \in G_k} \frac{a_t}{\sqrt{\sum_{s=1}^{t} a_s^2}}, \quad \tilde{S}_k := \sum_{t \in G_k} \frac{a_t}{\sqrt{\sum_{s=1}^{t_{k-1}} a_s^2}}. \tag{23}$$

By our precondition, we have $|S_k| \le M$ for all $k \in [K]$. Further, we have

$$|S_k - \tilde{S}_k| \le \sum_{t \in G_k} |a_t| \cdot \left( \frac{1}{\sqrt{\sum_{s=1}^{t_{k-1}} a_s^2}} - \frac{1}{\sqrt{\sum_{s=1}^{t} a_s^2}} \right) \le |G_k| \cdot \left( \frac{1}{\sqrt{\sum_{s=1}^{t_{k-1}} a_s^2}} - \frac{1}{\sqrt{\sum_{s=1}^{t_k} a_s^2}} \right) \tag{24}$$

$$\overset{(i)}{\le} L \cdot \frac{\sum_{t=t_{k-1}+1}^{t_k} a_s^2}{\left( \sum_{s=1}^{t_{k-1}} a_s^2 \right)^{3/2}} \overset{(ii)}{\le} L \cdot \frac{L}{2(\alpha^2(k-1)L)(\sum_{s=1}^{t_{k-1}} a_s^2)^{1/2}} = \frac{L}{2\alpha^2(k-1)} \cdot \frac{1}{\sqrt{\sum_{s=1}^{t_{k-1}} a_s^2}}, \tag{25}$$

where (i) uses the inequality $\frac{1}{\sqrt{x}} - \frac{1}{\sqrt{x+y}} \le \frac{y}{2x^{3/2}}$ for $x, y \ge 0$, and (ii) uses the bounds $\sum_{t_{k-1}+1}^{t_k} a_s^2 \le t_k - t_{k-1} \le L$ and $\sum_{s=1}^{t_{k-1}} a_s^2 \ge \alpha^2 t_{k-1} = \alpha^2(k-1)L$. By the triangle inequality, this implies that

$$|\tilde{S}_k| \le |S_k| + |S_k - \tilde{S}_k| \le M + \frac{L}{2\alpha^2(k-1)\sqrt{\sum_{s=1}^{t_{k-1}} a_s^2}}, \tag{26}$$

and thus for any $k \ge 2$ that

$$\left| \sum_{t \in G_k} a_t \right| = \left| \sum_{t \in G_k} \frac{a_t}{\sqrt{\sum_{s=1}^{t_{k-1}} a_s^2}} \right| \cdot \left| \sqrt{\sum_{s=1}^{t_{k-1}} a_s^2} \right| \le \left( M + \frac{L}{2\alpha^2(k-1) \cdot \sqrt{\sum_{s=1}^{t_{k-1}} a_s^2}} \right) \cdot \sqrt{\sum_{s=1}^{t_{k-1}} a_s^2} \tag{27}$$

$$\le M \sqrt{\sum_{s=1}^{t_{k-1}} a_s^2} + \frac{L}{2\alpha^2(k-1)} \le M\sqrt{(k-1)L} + \frac{L}{2\alpha^2(k-1)}. \tag{28}$$

For $k = 1$, we have trivially $\left| \sum_{t \in G_1} a_t \right| \le |G_1| \le L$. Summing the bounds over $k \in [K]$ yields

$$\left| \sum_{t=1}^{T} a_t \right| \le L + \sum_{k=1}^{K} \left| \sum_{t \in G_k} a_t \right| \le L + M\sqrt{L} \cdot \sum_{k=2}^{K} \sqrt{k-1} + \frac{L}{2\alpha^2} \sum_{k=2}^{K} \frac{1}{k-1} \tag{29}$$

$$\le L + \frac{2}{3} M\sqrt{L} K^{3/2} + \frac{L}{2\alpha^2} \log_2 K \le \lceil T^{\beta} \rceil + \frac{2}{3} M \sqrt{\lceil T^{\beta} \rceil} \cdot T^{(3(1-\beta))/2} + \frac{1}{2\alpha^2} \lceil T^{\beta} \rceil \log_2 \left( T^{1-\beta} \right) \tag{30}$$

$$\le 2T^{\beta} + 2MT^{3/2-\beta} + \frac{2}{\alpha^2} T^{\beta} \log T. \tag{31}$$

Choosing $\beta = 3/4$, we obtain

$$\left| \sum_{t=t_0+1}^{t_f} a_t \right| \leq 2(M + 1 + \log T/\alpha^2)T^{3/4}. \tag{32}$$

Dividing by $T$ on both sides yields the desired result.

$\square$

With Lemma F.4 and F.5, we can prove Theorem 2.

*Proof.* THe update mechanism for algorithm 3 can be simplified as:

$$\hat{s}_{t+1} = \hat{s}_t + \eta \frac{\text{err}_t - \alpha}{\sqrt{\sum_{s=1}^{t}(\text{err}_s - \alpha)^2}} = \hat{s}_1 + \eta \sum_{s=1}^{t} \frac{\text{err}_s - \alpha}{\sqrt{\sum_{i=1}^{t}(\text{err}_i - \alpha)^2}}. \tag{33}$$

Note that we have $\hat{s}_{t+1} \in [-\eta, D + \eta]$ for all $t \geq 0$ (Lemma F.1), which implies that

$$\left| \sum_{t=t_0+1}^{t_f} \frac{\text{err}_t - \alpha}{\sqrt{\sum_{s=1}^{t}(\text{err}_s - \alpha)^2}} \right| = \frac{1}{\eta}\left| \hat{s}_{t_f+1} - \hat{s}_{t_0+1} \right| \leq \frac{D + 2\eta}{\eta} \quad \text{for any} \quad 0 \leq t_0 < t_f. \tag{34}$$

Note that $|\text{err}_t - \alpha| \in [\alpha, 1]$ for all $t$. Therefore, we can invoke Lemma F.2 below with $a_t = \text{err}_t - \alpha$ and $M = (D + 2\eta)/\eta$ to obtain that for any $T \geq 1$,

$$\left| \frac{1}{T} \sum_{t=1}^{T} \text{err}_t - \alpha \right| \leq 2\left( \frac{D + 3\eta}{\eta} + \alpha^2 \log T \right) T^{-1/4} \leq \mathcal{O}(\alpha^{-2}T^{-1/4} \log T), \tag{35}$$

where the later bound holds for any $\eta = \Theta(D)$. This proves Theorem 4.1.

$\square$

### F.2 Negative Results for Naive Approaches

**Theorem F.3.** *Extension of Theorem 2.3 from Zaffran et al. (2023). Assume that the data are assgined the pre-defined weights $w_1, w_2, \ldots, w_{n+1}$, and that the method outputting $\widehat{C}_{n,\alpha}$ achieves group conditional coverage. Then, for any distribution $P \in \mathcal{P}$ and any mask $m \in \mathcal{M}$ such that $P_M(m) > 0$, the following holds:*

$$\mathbb{P}\left( \widehat{C}_{n,\alpha}(X, m) = \infty \right) \geq 1 - \alpha - \Delta_{m,n}$$

$$- \sum_{i=1}^{n} w_i d_{\text{TV}}(R(Z), R(Z^i)),$$

*where:* $\Delta_{m,n} = \sqrt{2\left(1 - \left(1 - \frac{P_M(m)^2}{2}\right)^{n+1}\right)}$, $w_i$ *are predefined weights assigned to the data point,* $Z = \{(X_1, y_1), (X_2, y_2), \ldots, (X_{n+1}, y_{n+1})\}$ *denotes the entire dataset,* $Z^i$ *is the same dataset as* $Z$, *except that the test point* $(X_{n+1}, y_{n+1})$ *is swapped with the $i$-th data point* $(X_i, y_i)$, $R(Z) = |y_i - f(X_i)|$ *represents the residual,* $d_{\text{TV}}(R(Z), R(Z^i))$ *is the total variation distance between the residual distributions from dataset* $Z$ *and dataset* $Z^i$.

**Lemma F.4.** *For $P$ and $Q$ two probability distributions, and $n \in \mathbb{N}^*$, it holds:*

$$TV(P^n, Q^n) \leq \sqrt{2\left(1 - \left(1 - \frac{TV(P, Q)^2}{2}\right)^n\right)}.$$

*Proof.* The proof of this lemma is based on the relationship between the total variation distance and the Hellinger distance between two probability distributions denoted by $H(\cdot, \cdot)$ (see Tsybakov & Tsybakov (2009)).

Let $n \in \mathbb{N}^*$ and let $P$ and $Q$ be two probability distributions.

On the one hand, note that:

$$TV(P,Q) \leq H(P,Q). \tag{36}$$

On the other hand, observe that:

$$H^2(P^n, Q^n) = 2\left(1 - \left(1 - \frac{H^2(P,Q)}{2}\right)^n\right). \tag{37}$$

Therefore, by combining Equations 36 and 37 (that can be found in Tsybakov & Tsybakov (2009)), we obtain the desired result. $\qquad\square$

**Lemma F.5.** *Let $W$ be a random variable such that $0 \leq W \leq 1$ and $\mathbb{E}[W] \geq \beta$ with $\beta \in [0,1]$. Then, for any $t > 0$, it holds $\mathbb{P}(W \geq 1 - t) \geq 1 - \frac{1-\beta}{t}$.*

*Proof.* Let $t > 0$. As $W \leq 1$, $1 - W \geq 0$. Therefore, using Markov's inequality:

$$\mathbb{P}(1 - W \geq t) \leq \frac{\mathbb{E}[1-W]}{t} = \frac{1 - \mathbb{E}[W]}{t} \leq \frac{1-\beta}{t}. \tag{38}$$

Noting that:

$$\mathbb{P}(1 - W \geq t) = \mathbb{P}(W \leq 1 - t) = 1 - \mathbb{P}(W \geq 1 - t), \tag{39}$$

we finally get $\mathbb{P}(W \geq 1 - t) \geq 1 - \frac{1-\beta}{t}$. $\qquad\square$

With Lemma F.4 and F.5, we can prove Theorem 1.

*Proof.* Let $n \in \mathbb{N}^*$ be the total training size (proper training and calibration). Let $\alpha \in [0,1]$. Let $\hat{C}_{n,\alpha}$ be a group conditional prediction interval. Let $P$ be a distribution on $\mathcal{X} \times \mathcal{M} \times \mathcal{Y}$. Let $m_0 \in \mathcal{M}$. Denote by $\rho := P_M(\{m_0\})$. Let $D > 0$.

Define $Q$ as another distribution on $\mathcal{X} \times \mathcal{M} \times \mathcal{Y}$ such that for any $A \subseteq \mathcal{X}$, for any $L \subseteq \mathcal{M}$ and for any $B \subseteq \mathcal{Y}$:

$$Q(A \times L \times B) := P(A \times L\backslash\{m_0\} \times B) + P_{(X,M)}(A \times \{m_0\})R(B), \tag{40}$$

with $R$ defined on $\mathcal{Y}$, uniform on $[-D, D]$.

Recall that the total variation distance between two probability distributions on $\mathcal{Z}$, say $P$ and $Q$, is defined as:

$$TV(P,Q) := \sup_{Z \in \mathcal{Z}} |P(Z) - Q(Z)|. \tag{41}$$

On the one hand, by construction, $TV(P,Q) \leq P_M(\{m_0\}) = \rho$. Hence, using Lemma F.4:

$$TV(P^{\otimes(n+1)}, Q^{\otimes(n+1)}) \leq \sqrt{2\left(1 - \left(1 - \frac{\rho^2}{2}\right)^{n+1}\right)}. \tag{42}$$

Therefore, for any $A \subseteq \mathcal{X}$, for any $L \subseteq \mathcal{M}$ and for any $B \subseteq \mathcal{Y}$:

$$P^{\otimes(n+1)}(A \times L \times B) \geq Q^{\otimes(n+1)}(A \times L \times B) - \sqrt{2\left(1 - \left(1 - \frac{\rho^2}{2}\right)^{n+1}\right)}. \tag{43}$$

On the other hand, the prediction intervals satisfy group conditional coverage:

$$1 - \alpha - \sum_{i=1}^{n} w_i d_{\mathrm{TV}}(R(Z), R(Z^i))$$

$$\leq \mathbb{P}_{Q^{\otimes(n+1)}} \left( Y^{(n+1)} \in \hat{C}_{n,\alpha}\left(X^{(n+1)}, m_0\right) \mid M^{(n+1)} = m_0 \right)$$

$$= \mathbb{E}_{Q^{\otimes(n+1)}} \left[ \mathbf{1}\left\{ Y^{(n+1)} \in \hat{C}_{n,\alpha}\left(X^{(n+1)}, m_0\right) \right\} \mid M^{(n+1)} = m_0 \right]$$

$$= \mathbb{E}_{Q^{\otimes(n)}} \left[ \mathbb{E}_Q \left[ \mathbf{1}\left\{ Y^{(n+1)} \in \hat{C}_{n,\alpha}\left(X^{(n+1)}, m_0\right) \right\} \mid M^{(n+1)} = m_0, \left(X^{(k)}, M^{(k)}, Y^{(k)}\right)_{k=1}^{n} \right] \right]$$

$$= \mathbb{E}_{Q^{(n)}} \left[ \mathbb{E}_Q \left[ \mathbb{E}_Q \left[ \mathbf{1}\left\{ Y^{(n+1)} \in \hat{C}_{n,\alpha}\left(X^{(n+1)}, m_0\right) \right\} \right. \right. \right.$$

$$\left. \mid X^{(n+1)}, M^{(n+1)} = m_0, \left(X^{(k)}, M^{(k)}, Y^{(k)}\right)_{k=1}^{n} \right] \mid M^{(n+1)} = m_0, \left(X^{(k)}, M^{(k)}, Y^{(k)}\right)_{k=1}^{n}$$

$$= \mathbb{E}_{Q^{\otimes(n)}} \left[ \mathbb{E}_Q \left[ \int_{\hat{C}_{n,\alpha}\left(X^{(n+1)}, m_0\right)} q\left(y \mid X^{(n+1)}, m_0\right) dy \mid M^{(n+1)} = m_0, \left(X^{(k)}, M^{(k)}, Y^{(k)}\right)_{k=1}^{n} \right] \right]$$

$$= \mathbb{E}_{Q^{\otimes(n)}} \left[ \mathbb{E}_Q \left[ \Lambda\left(\hat{C}_{n,\alpha}\left(X^{(n+1)}, m_0\right) \cap [-D, D]\right) \times \frac{1}{2D} \mid M^{(n+1)} = m_0, \left(X^{(k)}, M^{(k)}, Y^{(k)}\right)_{k=1}^{n} \right] \right]$$

$$= \mathbb{E}_{Q^{\otimes(n)}} \left[ \Lambda\left(\hat{C}_{n,\alpha}\left(X^{(n+1)}, m_0\right) \cap [-D, D]\right) \times \frac{1}{2D} \mid M^{(n+1)} = m_0 \right].$$

Note that $\Lambda\left(\hat{C}_{n,\alpha}\left(X^{(n+1)}, m_0\right) \cap [-D, D]\right) \times \frac{1}{2D} \leq 1$ almost surely. Therefore, using Lemma F.5, for any $t > 0$:

$$\mathbb{P}_{Q^{\otimes(n+1)}} \left( \Lambda\left(\hat{C}_{n,\alpha}\left(X^{(n+1)}, m_0\right) \cap [-D, D]\right) \times \frac{1}{2D} \geq 1 - t \right) \geq 1 - \frac{\alpha}{t}$$

$$\mathbb{P}_{Q^{\otimes(n+1)}} \left( \Lambda\left(\hat{C}_{n,\alpha}\left(X^{(n+1)}, m_0\right) \cap [-D, D]\right) \geq (1-t)2D \right) \geq 1 - \frac{\alpha}{t}$$

$$\Rightarrow \mathbb{P}_{Q^{\otimes(n+1)}} \left( \Lambda\left(\hat{C}_{n,\alpha}\left(X^{(n+1)}, m_0\right)\right) \geq (1-t)2D \right) \geq 1 - \frac{\alpha}{t}.$$

Let $t = 1 - \frac{1}{\sqrt{D}}$ and obtain $\mathbb{P}_{Q^{\otimes(n+1)}} \left( \Lambda\left(\hat{C}_{n,\alpha}\left(X^{(n+1)}, m_0\right)\right) \geq 2\sqrt{D} \right) \geq 1 - \frac{\alpha}{1 - \frac{1}{\sqrt{D}}}$.

Combining with Equation 43, we finally get:

$$\mathbb{P}_{P^{\otimes(n+1)}} \left( \Lambda\left(\hat{C}_{n,\alpha}\left(X^{(n+1)}, m_0\right)\right) \geq 2\sqrt{D} \right) \geq 1 - \frac{\alpha}{1 - \frac{1}{\sqrt{D}}} \geq \sqrt{2\left(1 - \left(1 - \frac{\rho^2}{2}\right)^{n+1}\right)}.$$

Letting $D \rightarrow +\infty$, the result is proven. $\qquad\square$

# G  ADDITIONAL EXPERIMENTS

## G.1  Comparison with Non-CP Methods

Given that this is the first CP work for time series with missing values, we used classical non-CP approaches commonly used for uncertainty quantification in time series prediction Blasco et al. (2024) including quantile regression, MC Dropout, and bayesian neural networks to ensure a meaningful evaluation. This incorporation aligns with prior work like Huang et al. (2024) which similarly incorporated these methods as baselines in their evaluations. Below we try to provide a clear explanation for these 3 baselines:

Quantile Regression (QR) (Koenker, 2005) is a powerful method for uncertainty quantification, as it directly models conditional quantiles of the target variable to estimate prediction intervals. For example, to generate a 90% prediction interval, QR models specific conditional quantiles of the response variable, such as the 5th and 95th percentiles. The model is trained by minimizing the pinball loss:

$$L_q(y, \hat{y}) = \begin{cases} q \cdot (y - \hat{y}), & \text{if } y \geq \hat{y}, \\ (1 - q) \cdot (\hat{y} - y), & \text{if } y < \hat{y}. \end{cases}$$

While QR does not inherently guarantee statistical coverage like CP, it serves as a classical and flexible baseline for evaluating uncertainty quantification methods. To adapt it for time series data in our setting, we use all previous data points as training data for each test instance to estimate the quantiles, thereby generating prediction intervals. This approach ensures a fair evaluation by maintaining the temporal order of the data.

MC Dropout (Gal & Ghahramani, 2016) is a technique used in neural networks to estimate uncertainty in predictions. Normally, dropout is a regularization method used during training to randomly turn off some neurons, helping the network generalize better. In MC dropout, by running the model multiple times with dropout enabled during evaluation, we generate a set of different predictions, which can be seen as samples from the model's uncertainty. To estimate the uncertainty, we look at the distribution of these predictions. In deployment, we generate $K$ predictions using MC Dropout and then calculate the 90% confidence intervals of the predicted values. Same as QR, we leverage all previous data points for training to ensure fair evaluation.

BNNs (Kendall & Gal, 2017) are a type of artificial neural network (ANN) that incorporates probabilistic reasoning into the network's architecture. In a traditional frequentist approach (e.g., standard neural networks), the parameters (weights and biases) are learned and then kept fixed. In contrast, a Bayesian approach treats the parameters as random variables with a probability distribution. The goal of training a BNN is to infer the posterior distribution of the parameters instead of finding the single optimal values. The posterior distribution represents all the information about the model parameters after observing the training data, taking into account both prior beliefs (the prior distribution) and the likelihood of the observed data. In deployment, the target label is typically modeled using a normal distribution. The model outputs two heads: one corresponding to the mean and the other to the log variance. To obtain the standard deviation, we compute the square root of the exponent of the log variance. The 90% prediction interval is then calculated as:

$$[\text{mean} - 1.645 \times \text{standard deviation}, \text{mean} + 1.645 \times \text{standard deviation}]$$

To adapt this method for time series data in our setting, we use all previous data points as the training set for each test instance to estimate the prediction intervals, ensuring a fair evaluation. The experimental results in Table 1 show that all these methods suffer from significant under-coverage, particularly QR and MC Dropout. This under-coverage can be attributed to the fact that QR often struggles with modeling the true conditional quantiles in the presence of high variability or model misspecification, while MC Dropout tends to underestimate uncertainty due to its reliance on a single model for approximating the posterior distribution. These results furthur highlight the effectiveness of CP methods.

## G.2 Coverage Guarantee under Frequentist Approach

The calculation of coverage in our work differs from the frequentist perspective, but aligns with recent work on CP for time series, such as Auer et al. (2023) and Sun & Yu (2023), where the empirical coverage rate is calculated as the percentage of test data points falling within the prediction intervals. This empirical measure serves as a practical approximation of frequentist coverage, particularly when the test data size is large. Additionally, we incorporate a frequentist perspective in our analysis by splitting the data into five distinct subsets and repeating the process across these splits. This methodology allows us to evaluate the variability in coverage and report the variance.

We also note that while asymptotic repetition can be computationally expensive, our approach is significantly more computationally efficient compared to other baselines. As mentioned in the main text, for the oil temperature dataset, our method completes tasks in approximately 5 minutes, whereas other methods take

over 20 minutes. This computational burden makes it infeasible to report coverage for other baselines, as detailed in Table 21.

In Table 12 , we present coverage results from a frequentist perspective, focusing on the oil temperature dataset under MNAR and MAR settings. The reported coverage is averaged over 100 trials, each consisting of 100 test data points. The results demonstrate that our method successfully maintains the frequentist coverage guarantee.

Table 12: Group conditional coverage of ACP-TSM ($\alpha = 0.1$). Each column represents a group categorized by the number of missing values.

|  |  | 0 | 1 | 2 | 3 | 4 | 5 |
|---|---|---|---|---|---|---|---|
| MCAR | Cov | $0.889\pm_{.00}$ | $0.904\pm_{.00}$ | $0.905\pm_{.00}$ | $0.906\pm_{.00}$ | $0.897\pm_{.00}$ | $0.898\pm_{.00}$ |
|  | Width | $24.36\pm_{.06}$ | $25.12\pm_{.06}$ | $25.23\pm_{.06}$ | $25.14\pm_{.06}$ | $25.79\pm_{.06}$ | $25.54\pm_{.06}$ |
| MAR | Cov | $0.900\pm_{.00}$ | $0.902\pm_{.00}$ | $0.904\pm_{.00}$ | $0.895\pm_{.00}$ | $0.905\pm_{.00}$ | $0.902\pm_{.00}$ |
|  | Width | $25.68\pm_{.06}$ | $26.32\pm_{.06}$ | $25.67\pm_{.06}$ | $26.45\pm_{.06}$ | $25.86\pm_{.06}$ | $25.34\pm_{.06}$ |

### G.3 Ablation Studies

We conduct additional ablation studies across both Temperature and Energy datasets under three missing data mechanisms. As shown in Tables 13 and 14, our method consistently achieves the smallest coverage gap across all subgroups. Furthermore, Table 15 presents an additional ablation study to validate the necessity of applying a second model. The results confirm that minimizing inefficiency loss leads to more efficient prediction intervals, while minimizing coverage loss enhances coverage guarantees.

Table 13: Ablation studies under different missing data mechanisms for Oil Temperature Data.

|  | MCAR | | | MAR | | | MNAR | | |
|---|---|---|---|---|---|---|---|---|---|
| Methods | Cov | Width | CovGap | Cov | Width | CovGap | Cov | Width | CovGap |
| Ours | 89.66 | 25.40 | 0.032 | 89.73 | 25.80 | 0.016 | 90.23 | 26.14 | 0.018 |
| w.o. Group Features | 89.23 | 25.38 | 0.083 | 89.62 | 26.12 | 0.036 | 89.45 | 26.02 | 0.057 |

Table 14: Ablation studies under different missing data mechanisms for Energy Data.

|  | MCAR | | | MAR | | | MNAR | | |
|---|---|---|---|---|---|---|---|---|---|
| Methods | Cov | Width | CovGap | Cov | Width | CovGap | Cov | Width | CovGap |
| Ours | 89.40 | 292.53 | 0.009 | 89.60 | 293.13 | 0.003 | 89.58 | 293.06 | 0.004 |
| w.o. Group Features | 89.56 | 292.86 | 0.005 | 89.42 | 293.05 | 0.006 | 89.47 | 293.15 | 0.006 |

### G.4 Generalization to large-scale time series data

To examine scalability, we have conducted additional experiments using large-scale financial data and stock exchange datasets sourced from Yahoo Finance, spanning multiple time horizons (1 month, 6 months, 1 year, and 5 years). Results presented in Table 16 demonstrate that our proposed method, ACP-TSM, robustly maintains the desired coverage rates while achieving the best prediction efficiency.

### G.5 Experiments on multi-step time series forecasting

We consider forecasting horizons of 5, 10, 25, and 50 steps during inference. We report results for these multi-step predictions using the Oil Temperature dataset under the MCAR missingness scenario in Table 17. Our results demonstrate that ACP-TSM consistently achieves the desired coverage guarantees while maintaining superior efficiency (narrowest prediction intervals) across different forecasting horizons.

Table 15: Ablation studies under different missing data mechanisms for Oil Temperature Data.

|  | MCAR | | | MAR | | | MNAR | | |
| Methods | Cov | Width | CovGap | Cov | Width | CovGap | Cov | Width | CovGap |
|---|---|---|---|---|---|---|---|---|---|
| Ours | 89.66 | 25.40 | 0.032 | 89.73 | 25.80 | 0.016 | 90.23 | 26.14 | 0.018 |
| w.o. coverage loss | 86.82 | 25.20 | 0.056 | 87.05 | 25.12 | 0.053 | 86.68 | 24.45 | 0.063 |
| w.o. inefficiency loss | 92.72 | 26.62 | 0.013 | 92.35 | 26.82 | 0.025 | 92.13 | 26.46 | 0.024 |

Table 16: Comparison of coverage rates and prediction interval widths for large scale high frequency financial time series data under MCAR.

| Method | 1M | | 6M | | 1Y | | 5Y | |
| | Coverage | Width | Coverage | Width | Coverage | Width | Coverage | Width |
|---|---|---|---|---|---|---|---|---|
| CP-MDA-Adaptive | 0.894 | 1.35 | 0.893 | 1.37 | 0.897 | 1.37 | 0.904 | 1.38 |
| MVSC | 0.902 | 4.12 | 0.898 | 4.42 | 0.903 | 4.56 | 0.907 | 4.72 |
| ACP-TSM | 0.895 | **1.28** | 0.902 | **1.32** | 0.899 | **1.33** | 0.898 | **1.33** |

### G.6 Comparisons with different SF-OGD Alternatives

In our current implementation, we adopt SF-OGD, though more recent advancements may offer performance gains. To further investigate this point, we conducted additional experiments replacing SF-OGD with ECI-integral and a decaying rate on three datasets under MCAR and MAR. As shown in Table 18, these alternatives yield improvements on certain datasets. However, the improvements are not consistent. We attribute this to two potential reasons. First, our method already incorporates a post hoc LSTM correction for coverage, which may reduce the benefit of additional online adjustments. Second, we believe that further tuning of parameters could further enhance performance, although this would introduce additional complexity.

More importantly, one notable advantage of SF-OGD is that its main tuning parameter: the maximum radius $D$, is easily interpretable as the maximum allowable deviation between the prediction and the ground truth, as discussed in Susmann et al. (2023). This parameter can be selected easily with the calibration set. In our method, SF-OGD is applied in an online and adaptive manner directly on the test set without requiring a separate tuning set. In contrast, hyperparameter tuning would require an additional validation set, increasing both computational burden and implementation complexity.

### G.7 Comparisons with Reweighting Alternatives.

More sophisticated reweighting schemes have been proposed to better capture temporal dependencies in time series, often by leveraging similarity measures or kernel-based weighting strategies. To furthur investigate this problem, we propose additional experiments over KOWCPI method (Lee et al., 2025). Specifically, we compare the comparison of our method (initial interval provide by KOWCPI) and CP-MDA-Exact+KOWCPI over Oil Temperature Data under MCAR and MAR. Results in Table 19 demonstrate that our method can still consistently improves efficiency while maintaining valid coverage guarantees. Additionally, Table 16 in Appendix G.8 supports our claim that our method outperforms the combination of CP-MDA-Exact and SPCI (Xu & Xie, 2023), a representative approach in conformal time series. We attribute this improvement to the design of inefficiency loss, while reweighting alternatives don't explicitly target at prediction efficiency. Additionally, these advanced reweighting techniques introduce significant computational overhead, which can undermine their practical effectiveness. Moreover, they often rely on additional assumptions to achieve asymptotic coverage, and their theoretical bounds typically include residual error terms. In contrast, a key advantage of ACP-TSM is its computational efficiency, and our design already includes a correction term to account for distribution shifts. Therefore, we argue that our current approach offers a favorable balance between accuracy and efficiency. We will include a detailed discussion of this comparison in the appendix in our revised version.

Table 17: Comparison of coverage rates and prediction interval widths for multi-step forecasting tasks on the Oil Temperature dataset under the MCAR missingness mechanism. Bold values indicate the best (smallest) prediction interval width.

| Method | 5 steps | | 10 steps | | 25 steps | | 50 steps | |
|---|---|---|---|---|---|---|---|---|
| | Coverage | Width | Coverage | Width | Coverage | Width | Coverage | Width |
| CP-MDA-Adaptive | 0.891 | 28.63 | 0.901 | 28.52 | 0.896 | 28.32 | 0.903 | 29.03 |
| MVSC | 0.894 | 33.15 | 0.898 | 33.46 | 0.902 | 34.67 | 0.897 | 33.75 |
| ACP-TSM | 0.891 | **25.90** | 0.892 | **25.96** | 0.892 | **25.81** | 0.889 | **25.58** |

Table 18: Performance of the compared methods for the three benchmark datasets with miscoverage rate $\alpha = 0.1$ under MCAR and MAR.

| | | Dataset | | | | | |
|---|---|---|---|---|---|---|---|
| Methods | | Elec | | OT | | Energy | |
| | | MCAR | MAR | MCAR | MAR | MCAR | MAR |
| ACP-TSM+decaying | Cov | 0.893 | 0.899 | 0.896 | 0.910 | 0.902 | 0.905 |
| | Width | 0.605 | **0.602** | **26.42** | 26.64 | 302.80 | 307.20 |
| | CovGap | 0.014 | **0.005** | 0.015 | 0.006 | 0.013 | 0.016 |
| ACP-TSM+ECI | Cov | 0.892 | 0.895 | 0.900 | 0.897 | 0.892 | 0.895 |
| | Width | **0.603** | 0.604 | 26.58 | **26.62** | 297.72 | **297.45** |
| | CovGap | 0.012 | 0.012 | 0.016 | 0.022 | 0.012 | 0.008 |
| ACP-TSM | Cov | 0.899 | 0.903 | 0.905 | 0.899 | 0.900 | 0.901 |
| | Width | 0.605 | 0.603 | 26.80 | 26.92 | **297.68** | 299.23 |
| | CovGap | **0.008** | 0.007 | **0.006** | **0.005** | **0.009** | **0.007** |

Table 19: Group conditional coverage of the compared CP algorithms ($\alpha = 0.1$). Each column represents a group categorized by the number of missing values.

| | | | 0 | 1 | 2 | 3 | 4 | 5 |
|---|---|---|---|---|---|---|---|---|
| MCAR | CP-MDA-Exact+KOWCPI | Cov | 0.959 | 0.910 | 0.918 | 0.900 | 0.883 | 0.872 |
| | | Width | 28.20 | 28.59 | 28.71 | 28.31 | 27.45 | 28.06 |
| | ACP-TSM | Cov | 0.894 | 0.892 | 0.896 | 0.902 | 0.898 | 0.904 |
| | | Width | **25.32** | **25.78** | **25.62** | **25.77** | **26.12** | **25.64** |
| MAR | CP-MDA-Exact+KOWCPI | Cov | 0.923 | 0.912 | 0.909 | 0.901 | 0.888 | 0.872 |
| | | Width | 27.29 | 28.81 | 28.66 | 28.24 | 27.62 | 27.96 |
| | ACP-TSM | Cov | 0.896 | 0.896 | 0.903 | 0.898 | 0.904 | 0.902 |
| | | Width | **26.48** | **25.62** | **25.72** | **25.53** | **25.82** | **25.97** |

### G.8 Comparisons with Conformal Prediction Methods for Time Series

We compare our method against combination of CP-MDA with SPCI Xu & Xie (2023). As shown in Table 20, experiments are conducted on three benchmark datasets. In particular, ACP-TSM consistently achieves the narrowest prediction intervals while maintaining valid coverage.

### G.9 Datasets Details

Datasets from four different domains are used in the experiments. Details are summarized below.

- **Electricity.** The Electricity Demand Forecasting dataset, introduced by Harries in 1999 (Harries et al. (1999)), tracks electricity usage and pricing in New South Wales and Victoria, Australia, with

Table 20: Performance of the compared methods for the three benchmark datasets with miscoverage rate $\alpha$ =0.1, under two missing mechanisms. Results were obtained by 5 repetitions with different seeds. Bold numbers indicate the best result, while significant under-coverage is canceled out.

| Methods | | Elec | | OT | | Energy | |
| --- | --- | --- | --- | --- | --- | --- | --- |
| | | MCAR | MAR | MCAR | MAR | MCAR | MAR |
| CP-MDA-Adaptive | Cov | $0.895\pm_{.0}$ | $0.895\pm_{.0}$ | $0.894\pm_{.0}$ | $0.896\pm_{.0}$ | $0.896\pm_{.0}$ | $0.896\pm_{.0}$ |
| | Width | $0.605\pm_{.0}$ | $0.604\pm_{.0}$ | $28.44\pm_{.12}$ | $28.42\pm_{.12}$ | $296.43\pm_{5.52}$ | $296.62\pm_{5.63}$ |
| | CovGap | $0.007\pm_{.0}$ | $0.007\pm_{.0}$ | $0.011\pm_{.0}$ | $0.018\pm_{.0}$ | $0.008\pm_{.0}$ | $0.008\pm_{.0}$ |
| MVSC | Cov | $0.898\pm_{.0}$ | $0.893\pm_{.0}$ | $0.897\pm_{.0}$ | $0.909\pm_{.0}$ | $0.902\pm_{.0}$ | $0.902\pm_{.0}$ |
| | Width | $0.631\pm_{.0}$ | $0.628\pm_{.0}$ | $30.67\pm_{.18}$ | $34.34\pm_{.18}$ | $576.33\pm_{15.6}$ | $523.74\pm_{15.8}$ |
| | CovGap | $0.003\pm_{.0}$ | $0.010\pm_{.0}$ | $\mathbf{0.002}\pm_{.0}$ | $0.231\pm_{.0}$ | $\mathbf{0.005}\pm_{.0}$ | $0.005\pm_{.0}$ |
| CP-MDA-SPCI | Cov | $0.896\pm_{.0}$ | $0.897\pm_{.0}$ | $0.897\pm_{.0}$ | $0.903\pm_{.0}$ | $0.892\pm_{.0}$ | $0.904\pm_{.0}$ |
| | Width | $0.597\pm_{.0}$ | $0.598\pm_{.0}$ | $27.62\pm_{.14}$ | $27.64\pm_{.14}$ | $298.15\pm_{4.03}$ | $298.73\pm_{4.11}$ |
| | CovGap | $0.003\pm_{.0}$ | $0.008\pm_{.0}$ | $0.006\pm_{.0}$ | $\mathbf{0.006}\pm_{.0}$ | $0.012\pm_{.0}$ | $0.008\pm_{.0}$ |
| ACP-TSM | Cov | $0.899\pm_{.0}$ | $0.900\pm_{.0}$ | $0.902\pm_{.0}$ | $0.890\pm_{.0}$ | $0.894\pm_{.0}$ | $0.896\pm_{.0}$ |
| | Width | $0.592\pm_{.0}$ | $\mathbf{0.588}\pm_{.0}$ | $\mathbf{25.71}\pm_{.04}$ | $\mathbf{25.80}\pm_{.04}$ | $\mathbf{292.53}\pm_{4.32}$ | $\mathbf{293.13}\pm_{4.30}$ |
| | CovGap | $\mathbf{0.002}\pm_{.0}$ | $\mathbf{0.004}\pm_{.0}$ | $0.011\pm_{.0}$ | $0.016\pm_{.0}$ | $0.009\pm_{.0}$ | $\mathbf{0.003}\pm_{.0}$ |

half-hourly data from May 7, 1996, to December 5, 1998. For our experiment, we focus on four key variables: nswprice and vicprice (electricity prices in New South Wales and Victoria) and nswdemand and vicdemand (electricity demand in each state). Our response variable, transfer, measures the electricity transferred between the two states. We select data from 9:00 AM to 12:00 PM to reduce daily fluctuations and discard an initial period with constant transfer values, resulting in 3,444 time points for analysis.

- **Oil Temperature.** The oil temperature data is collected by Zhou et al. (2021). The dataset consists of two years of recorded data from a region in a province of China. The data points are captured at one-minute intervals, resulting in a total of 70,080 data points. Each data point consists of 8 features, including the date of the point, the predictive value "oil temperature", and 6 different types of external power load features.

- **Appliances Energy.** The Appliances Energy Data set is introduced by the UCI Machine Learning Repository in Candanedo (2017). The data spans approximately 4.5 months with measurements taken every 10 minutes. House temperature and humidity conditions were monitored using a ZigBee wireless sensor network, where each wireless node transmitted data on temperature and humidity approximately every 3.3 minutes. Energy consumption data was logged every 10 minutes using M-Bus energy meters. Additionally, weather data from the nearest airport weather station (Chievres Airport, Belgium) was obtained from a public dataset provided by Reliable Prognosis (rp5.ru) and merged with the experimental datasets based on the date and time columns. Two random variables have been included in the dataset to test the regression models and filter out non-predictive attributes.

- **Air Quality.** AirQualityUCI is a real-world air pollution dataset from the UCI Machine Learning Repository. It contains 9,358 hourly averaged observations collected from an array of five metal-oxide chemical sensors deployed in an Italian city, together with reference measurements from a certified analyzer. Unlike the other datasets, missingness in this dataset arises naturally rather than being synthetically introduced. Following preprocessing, we remove instances for which the response variable CO(GT) is missing, as well as instances for which all covariates are missing, leaving 7,674 samples for model fitting. Across the covariates, there are 10,583 missing entries in total.

### G.10 Computational Efficiency

Experiments were conducted with 11th Gen Intel® Core™ i7-1165G7 processor (4 cores, 8 threads, base frequency 2.80GHz, turbo up to 4.70GHz). We report the runtimes of three major methods that can achieve

empirical marginal coverage across various datasets with MCAR conditions. The missingness is introduced synthetically in the same manner for all methods so we only compare the results on this situation. Our method consistently demonstrates superior performance while achieving significantly shorter runtimes across diverse scenarios, highlighting its practical advantages. Methods such as CP-MDA-Adaptive and MVSC rely on group-level calibration defined by missingness patterns. As the dimensionality increases, the number of possible missingness-pattern groups grows exponentially, which substantially increases computational cost and can limit practical applicability. In contrast, our method calibrates using the full calibration set without enumerating missingness-pattern groups, so its computational cost scales approximately linearly with dimensionality, making it more suitable for higher-dimensional settings such as the Energy dataset.

Table 21: Runtime of 3 major methods with different datasets for a single trial.

| Runtime | Dim of Input | Our method | CP-MDA-Adaptive | MVSC |
|---------|--------------|------------|-----------------|--------|
| Elec | 4 | 5s | 30s | 4mins |
| Oil | 6 | 5mins | 20mins | 25mins |
| Energy | 27 | 40mins | 16h | 22h |

### G.11 Neural Network Architecture

The neural network architecture used in this study is a bidirectional LSTM. The input dimension is based on the feature size of the calibration training data along with 7 additional supplementary inputs. The bidirectional LSTM is designed to capture information from both forward and backward directions in the sequence, which doubles the size of its hidden state output. In our implementation, the hidden state size is set to 64, and the model uses 2 LSTM layers. The output of the LSTM is then passed through two fully connected layers: the first reduces the 128-dimensional bidirectional LSTM output to 64 dimensions, and the second maps this to 2 final output dimensions. To reduce the risk of overfitting, dropout regularization with a rate of 0.5 is applied after the first fully connected layer. For training, the model utilizes the Adam optimizer with a learning rate of $lr = 10^{-5}$, and the batch size is set to 32 to balance computational efficiency and learning stability.

Our procedure is strictly rolling in both training and inference. The correction applied at time $t$ is computed using only information available up to time $t$ (i.e., past covariates/masks and past nonconformity information), and it never uses future observations or future labels. Although we implement the correction network as a bidirectional LSTM for convenience, the bidirectionality is applied only within the historical window befor $t$ when forming a representation for time $t$; the input sequence does not include any points after $t$. At test time, the refined interval for time $t$ depends only on data observed up to $t$; in particular, no future features (or any future labels) are accessed when producing the correction.

## H    Practical Deployment and Maintenance Under Distribution Shift

We briefly discuss how to deploy and maintain our method in real-world time-series applications, where distribution shift is common. At a high level, the method consists of (i) a base correction model (e.g., an LSTM) frozen and test time and (ii) a lightweight correction and calibration layer that adjusts uncertainty estimates using the most recent data (adaptive fine-tuning).

**Online updating of the correction model.**    In deployment, we recommend updating the correction (calibration) component online or in frequent mini-batches as new observations arrive. This update is computationally inexpensive relative to retraining the base forecaster, and it directly targets coverage control under evolving noise levels and distribution shift. In our experiments, this correction mechanism is sufficient to maintain stable performance even when the test-time distribution changes noticeably. For example, on the Oil Temperature dataset, Figure 6 indicates substantial shift during the test period, yet our method remains stable, which we attribute primarily to the adaptive fine-tuning component that continuously leverages recent observations to adjust the prediction procedure.

**When to retrain the base LSTM.** In contrast, frequent retraining of the base LSTM is not always necessary. We view retraining as an occasional maintenance step that can be triggered when (1) sufficient new data have accumulated, or (2) standard CP metrics (e.g., coverage) degrade persistently. This separation, frequent lightweight correction updates versus occasional base-model retraining, provides a practical balance between robustness and computational cost.

**Multi-step forecasting.** The same protocol applies to multi-step forecasting: the base correction model can remain fixed for extended periods, while the correction component continues to adapt to recent data. Empirically, Table 16 shows that our method remains effective in the multi-step setting, suggesting that the correction layer can track evolving uncertainty even when prediction horizons increase.

Overall, this deployment protocol highlights a practical tradeoff. More frequent retraining of the correction model may improve predictive accuracy under severe shifts, but it increases computational overhead. Our default recommendation is therefore to update the correction component regularly, and retrain the base LSTM only when monitoring signals indicate sustained performance degradation.

