# OpenReview forum: "Post-Training Adaptive Conformal Prediction for Incomplete Time Series"
_TMLR — Accepted by TMLR_

### Review · Reviewer_PgAr · 2026-01-26

**Summary Of Contributions:**

This paper introduces ACP-TSM, a model-agnostic post-hoc method to improve Conformal Prediction (CP) for time series with missing covariates. The authors leverage an LSTM with a dual loss function to optimize both group-conditional coverage and interval efficiency. By incorporating temporal context and missingness-aware features, the method handles non-exchangeable data more effectively than existing approaches. Results across MCAR, MAR, and MNAR mechanisms demonstrate that the approach produces tighter prediction intervals while preserving valid coverage with minimal computational overhead.

**Additional Comments:**

The problem addressed is interesting, and the paper appears to provide an initial solution for a specific context. The paper could be revised in terms of presentation to improve clarity and to develop a stronger argumentation.

**Audience:**

Yes

**Audience Explanation:**

Uncertainty quantification for time series with missing data is a very specific topic, but it remains interesting and the approach could, according to the authors, be adapted to other contexts.

**Claims And Evidence:**

No

**Claims Explanation:**

The paper is very difficult to read. The problem statement and the proposed solution are not very clear. Although the method seems to achieve good results in the experiments, it does not appear to be particularly novel or relevant; rather, it seems overly complex due to the large number of parameters, multiple loss components, sampling strategies, trade-offs, and so on. Some of the claimed properties do not seem to result from the method itself but rather from the use of the LSTM. The experiments are fairly well conducted (with multiple datasets, various contexts, a k‑fold approach, sensitivity analyses, and comparisons with recent competitors), but the methodology suggests that the work is based more on a loss function designed to minimize a criterion than on the introduction of a genuinely new method.

**Requested Changes:**

Here are some suggestions and questions (not necessarily leading to changes):
- The paper is quite difficult to understand and read, especially for readers who are not experts in the field. It would be very helpful to introduce or revise some key points, such as the issue of *non-exchangeability*, the concept of *Conformal Prediction*, the main objectives, and how the proposed approach addresses them. It would also be important to better define the scope in which this method applies.  A scheme of the proposed methodology would also be beneficial.
- “(NN)” in the abstract is unnecessary; it would be better to provide the actual acronym of the algorithm instead.
- **Introduction:** Add *“to our knowledge”* to the sentence *“Our work is the first to explore CP in time series data with missing covariates.”*
- *“Our work is the first to explore CP in time series data with missing covariates”*, does this refer to a supervised framework?
- The last lines of the *Related Work* section actually describe contributions. They should be moved to another section (e.g., Section 4 or 5.2).
- Sections 3 and 4 could be merged.
- The concept of *Conformal Prediction* is poorly presented. What is the main principle behind this methodology? What are its goals and issues? And how does the proposed approach address them? This concept should be presented at a higher level, with more perspective, so that it is understandable to a broader audience.
- **Section 4:** “The goal is to construct a prediction…”, should this be at time $t + k$ rather than $t + 1$?
- **Section 5.1:** It is unclear what a *calibration set* refers to. It would be beneficial to first present the main principles of the method rather than immediately describing it through mathematical notation.
- **Section 5.2:**
  - $C_n$  is not defined. Is it the prediction interval depending on $X_i$?
  - If a group $G_m$ has only a few observations and might not be representative, what ensures that the model does not overfit this missing pattern?
  - *lower_coverage* and *upper_coverage* are not very useful; a simple notation like $[ \ldots ; \ldots ]$ would suffice.
  - How do we move from Equation (4) to (7)? How is the product of the bounds derived?
  - How do the loss functions allow the method to satisfy or optimize *Marginal Coverage*?
  - Figure 2 is not very clear: the subfigures are not labeled and appear to be identical.
  - All notations should be clearly defined, e.g., $L_t$ and $U_t$.
  - In Equation (13), the loss is not complete since it lacks the $L_{cov}$ component.
  - The section *Post-training LSTM with Dual Loss* is not clear; it would be helpful to add a diagram.
- How could the approach be adapted for multiple imputation?
- **Tables 2 and 3:** The method performs better in width but not always in coverage.
- **Experiments:** There are many tables; could some be replaced by figures for better readability?
- **Tables 9 and 10:** The results appear very stable across different penalty values, which is quite surprising. How do you explain this? Would an analysis of the loss and its components help better understand this behavior?
- **Conclusion:** What are the main limitations and drawbacks of the proposed approach?

---

> ### Author Response · Authors · 2026-02-27
>
> We thank the reviewers for their careful reading of our manuscript and for providing constructive feedback. We have revised the paper accordingly, and all changes are highlighted in blue. Below, we respond to each comment point by point.
>
> **Q1 and Q7: Presentation of Conformal Prediction Method**
>
> Thank you for pointing this out. We agree that our original presentation of conformal prediction  was too technical for readers who are not experts in the area and did not provide sufficient high-level intuition. We have substantially revised the Background section to improve accessibility for a broader audience. In the revision, we (i) introduce the central principle of CP at a high level: using calibration residuals/nonconformity scores to turn any predictor into prediction intervals/sets with finite-sample statistical  guarantees; (ii) clearly state CP’s goals (distribution-free uncertainty quantification with coverage guarantees) as well as common practical limitations (e.g., reliance on exchangeability and the possibility of miscoverage under distribution shift or temporal dependence); and (iii) add a dedicated subsection discussing non-exchangeability.
>
> We also more formally define the problem setting in Section 3. In addition, at the beginning of Section 4, we provide a high-level overview of the proposed methodology, explain how it addresses these challenges, and include a schematic diagram of the overall procedure. We hope these changes make our paper more accessible to a broader audience, and we appreciate your feedback on our revision.
>
> **Q2: Inappropriate Acronym “(NN)” in the Abstract.**
>
> Thank you for pointing this out. We have removed “(NN)” from the abstract.
>
> **Q3: Refining the “First” Claim**
>
> Thank you for the suggestion. We have revised the sentence to “To our knowledge, our work is the first to explore CP in the setting of time series data with missing covariates.” in the Introduction section.
>
> **Q4: Clarification on “Our work is the first to explore CP in time series data with missing covariates”**
>
> Thank you for the comment. We agree that our original wording could be interpreted too broadly. Our intended claim is restricted to the joint setting of conformal prediction for time series where covariates contain missing values, and to our best knowledge this specific combination has not been explicitly addressed in prior literature. To avoid any overstatement, we have revised the Introduction to make the scope explicit and more cautious. Specifically, we changed the sentence to: “To our knowledge, we are the first to explore CP in the setting of time series data with missing covariates.”
>
> **Q5: Removal of the last lines of the Related Work**
>
> Thank you for pointing this out. We have moved the last sentence of the Related Work section to the beginning of Section 4 to more clearly highlight the contribution of our proposed methodology.
>
> **Q6: Sections 3 and 4 could be merged.**
>
> We thank the reviewer for this valuable suggestion. In response, we merged the original Sections 3 and 4 into a single section titled “Preliminaries and Problem Setup.” This consolidated section first introduces missingness mechanisms and conformal prediction (along with relevant variants), and then presents our problem formulation and setup.
>
> **Q8: Section 4 time index $t+k$.**
>
> Thank you for pointing this out. We revised this part to clarify that we predict over a predefined forecasting horizon (rather than only the next time point). Specifically, we now write:
> ''Let $k \in \{1,\dots,K\}$ denote the forecasting horizon, where $K \in \mathbb{N}_+$ is a user-specified maximum horizon.''
>
> **Q9: Clarification on \textit{calibration set} in Sec 5.1**
>
> We thank the reviewer for pointing out this important issue. Here, the calibration set refers to the set of data points used to compute nonconformity scores and determine the quantile threshold for constructing a prediction interval for a new test point. We agree that our original description did not clearly convey the main principle. We therefore revised the first paragraph of Section 4.1 to first explain how prior approaches handle missingness: they select a subset of calibration data that matches the test point’s missingness pattern, so as to mitigate the non-exchangeability induced by missing covariates.
>
> **Q10: Definition of $C_n$.**
>
> Thank you for pointing this out. Here, $C_n$ denotes the prediction interval constructed conditional on $X_i$ and $M_i$. We have revised this part to explicitly define $C_n$ and clarify its dependence on these quantities.

---

> > ### Author Response · Authors · 2026-02-27
> >
> > **Q11: Overfitting for less representative groups**
> >
> > Thank you for raising this concern. We indeed feed hand-engineered missingness-pattern features into the predictor, but the missingness pattern is only one component of the input rather than a standalone identifier. To mitigate the risk that the model “memorizes” an under-represented pattern, we apply dropout during training, which randomly perturbs the effective feature set and prevents the predictor from consistently relying on any single missingness-pattern signal. In addition, our procedure includes an adaptive stabilization mechanism that reduces sensitivity to sparsely observed patterns and helps avoid high-variance behavior when a pattern group is small. Taken together, the shared training objective, dropout regularization, and adaptive stabilization ensure that incorporating missingness-pattern inputs does not lead to overfitting to rare, potentially unrepresentative patterns. Empirically, Figures 5-6 in our manuscript show that across all missingness scenarios, our method recovers the desired coverage.
> >
> > **Q12: Definition of lower\_coverage and upper\_coverage**
> >
> > Thank you for pointing this out. We introduce lower\_coverage and upper\_coverage to define a differentiable conditional-coverage loss for learning. If we directly use the standard coverage indicator $\mathbb{1}\{Y \in C_n(\cdot)\}$, the loss is step-wise constant and discontinuous (and thus non-differentiable), which makes gradient-based optimization unstable or infeasible. Our lower\_coverage and upper\_coverage terms provide a smooth surrogate that captures violations on the lower and upper sides of the interval, enabling effective training while still targeting conditional coverage.
> >
> > **Q13: Clarification from Equations 4 to 7**
> >
> > In our revision, Eqs 4 to 7 in our submission become Eqs. 7 to 11. Eq. (7) measures, for each missingness group, the deviation between the group-wise (conditional) empirical coverage and the target level $1-\alpha$, and our goal is to make this deviation as small as possible. However, the empirical coverage in Eq. (7) is computed using the hard indicator $\mathbf{1}\{Y_i \in C_n(\cdot)\}$, i.e., the fraction of test points in the group whose outcomes fall inside the prediction interval, which is discontinuous and therefore not suitable for gradient-based optimization. To obtain a smooth objective, we replace this indicator with a sigmoid-based surrogate. Since the sigmoid $\sigma(\cdot)$ is bounded in $[0,1]$ and approaches $1$ as its input goes to $+\infty$ and $0$ as its input goes to $-\infty$, we can approximate the two one-sided conditions defining interval inclusion: $Y_i \ge L_i$ and $Y_i \le U_i$. Specifically, when $Y_i\in[L_i,U_i]$, both margins $Y_i-L_i$ and $U_i-Y_i$ are positive; with a sufficiently large smoothing parameter $c$, $\sigma(c(Y_i-L_i))\approx 1$ and $\sigma(c(U_i-Y_i))\approx 1$, so their product closely matches the hard indicator. Conversely, if $Y_i$ falls outside the interval, at least one margin becomes negative, driving the corresponding sigmoid term toward $0$, and thus the product $\sigma(c(Y_i-L_i))\,\sigma(c(U_i-Y_i))$ becomes close to $0$. Averaging this smooth surrogate over $i\in G_m$ yields the smooth empirical coverage in Eq. (7).

---

> > > ### Author Response · Authors · 2026-02-27
> > >
> > > **Q14: How does the loss function enforce marginal coverage**
> > >
> > > We clarify that marginal coverage is a direct consequence of our group-conditional coverage guarantee.
> > > Let $\{G_m\}\_{m=1}^M$ denote the partition of the sample space induced by the missingness patterns, and let $M^{(t+k)}\in\{1,\ldots,M\}$ be the corresponding pattern label.
> > > If our method guarantees, for every $m\in\{1,\ldots,M\}$,
> > > $$
> > > P(Y^{(t+k)} \in C_\alpha (X^{(t+k)}, M^{(t+k)})|M^{(t+k)} = m) \ge 1-\alpha.
> > > $$
> > > then marginal coverage follows immediately from the law of total probability:
> > > $$
> > > P( Y^{(t+k)} \in C_\alpha(X^{(t+k)}, M^{(t+k)}))
> > > = \sum_{m=1}^M P( Y^{(t+k)} \in C_\alpha(X^{(t+k)}, m)\|M^{(t+k)} = m )P(M^{(t+k)}=m)
> > > \ge 1-\alpha.
> > > $$
> > > Therefore, our loss functions are not used to directly enforce marginal coverage; rather, they are designed to encourage satisfaction of the stronger group-conditional constraints while improving efficiency, i.e., producing tighter prediction intervals. We thanks the reviewer for raising this important point and add this clarification in Section 3 in our revision.
> > >
> > > **Q15: Clarification on Figure 2**
> > >
> > > Thank you for pointing this out. We agree that Figure 2 in our submission was not presented clearly. In the revision, we zoom in to shorten the x-axis and add subcaptions to each subfigure. The revised figure shows the same qualitative trends: for consecutive time points, the resulting prediction intervals have similar lengths, indicating comparable uncertainty, which motivates our temporal adjustment approach. Moreover, comparing the electricity dataset (top) and the oil temperature dataset (bottom), we observe that the oil temperature series is more scattered. This is consistent with the larger distribution shifts and higher uncertainty observed for oil temperature in our experiments (see Figure 6 and Table 1 in our revised manuscript).
> > >
> > > **Q16: Definitions of Notations**
> > >
> > > We thank the reviewer for pointing this out. We have revised the notation and definitions throughout the paper.
> > >
> > > **Q17: Clarification of Eq. (13)**
> > >
> > > We thank the reviewer for pointing out this important point. Our original intention in Eq. (17) was to define the loss used in the efficiency-aware boosting module, which is designed to quantify interval efficiency (i.e., to encourage shorter prediction intervals). This term is not meant to be the full training objective on its own. We agree that it is more appropriate to also incorporate a coverage-related loss, and we have revised the objective accordingly by adding this term.
> > >
> > > **Q18: Diagram for methodology section**
> > >
> > > We thank the reviewer for this important comment. We agree that the presentation in Section 4 can be improved. In the revised manuscript, we add a schematic diagram to illustrate the overall pipeline and the two loss components, and we provide a clearer step-by-step algorithmic description to better explain the ACP-TSM procedure (Figure 3).
> > >
> > > **Q19: Adaptation for Multiple Imputation**
> > >
> > > Our method can be naturally extended to multiple imputation by treating the imputation procedure as a mapping from incomplete to complete data and then applying conformal prediction post hoc. Concretely, suppose we generate $B$ completed versions of each sample via multiple imputation, yielding $\{X^{(b)}\}_{b=1}^B$. We can then (i) form a single prediction set by aggregating the $B$ imputation-specific prediction sets (e.g., take the union for conservativeness, or use an aggregated score such as the average/quantile of conformity scores across imputations) and calibrate the aggregation on the calibration set, or (ii) incorporate imputation uncertainty directly by defining the conformity score as an aggregate across imputations (e.g., the maximum or a high quantile over $b$) and performing conformal calibration on this aggregated score. In both cases, the base predictor is still trained once (on training data), while calibration is performed on a held-out calibration set, so validity is obtained through the standard conformal guarantee under exchangeability of the aggregated scores. More broadly, since conformal prediction is model-agnostic and applied as a post-hoc uncertainty quantification step, it can accommodate multiple imputation as long as the imputation procedure is fixed (trained on the training set) and applied consistently to calibration and test points.

---

> > > > ### Author Response · Authors · 2026-02-27
> > > >
> > > > **Q20: Tables 2 and 3: The method performs better in width but not always in coverage.**
> > > >
> > > > We agree that our method does not always achieve the best empirical coverage in every setting, but it remains competitive overall. To provide a more complete picture beyond marginal coverage alone, we additionally report the Average Coverage Gap in Table 1. This metric is defined as the mean, across groups, of the absolute deviations of group wise coverage from the target level, and it quantifies how far group conditional coverage departs from the desired coverage, averaged over all missingness pattern groups.
> > > > Additionally, our objective is calibration to the target coverage rather than “the higher the better”: coverage substantially above the target typically reflects overly conservative (wider) intervals, so the relevant criterion is closeness to the target, not maximal coverage.
> > > >
> > > > Among methods that achieve marginal coverage, our approach attains comparable and in several cases smaller coverage gaps, while producing substantially narrower prediction intervals. In contrast, methods such as MVSC can be competitive in coverage in some settings, but typically do so with significantly wider intervals. Finally, our method is substantially more computationally efficient, making it a practical solution.
> > > >
> > > > **Q21: Presentation of Experimental Results (Tables vs. Figures)**
> > > >
> > > > Thank you for this helpful suggestion. We agree that an excessive number of tables can reduce readability. In the revised manuscript, we replace several tables with figures that more clearly highlight the main trends (e.g., Tables 2, 3, and 5 are now presented as Figures 4, 5, and 7). We expect this reorganization to make the empirical results easier to scan, and we welcome any further feedback on the revised presentation.
> > > >
> > > > **Q22: Sensitivity to Hyperparameters**
> > > >
> > > > We thank the reviewer for pointing out this important point. The results are not uniformly robust across all settings: for example, in Table 9 on the Energy dataset, when $\lambda_{\mathrm{reg}}=100$ the performance drops noticeably, indicating that overly strong regularization can indeed harm the fit. Regarding hyperparameter choices, $c$ is relatively easy to tune because it only controls the smoothness of the sigmoid approximation to the indicator; once $c$ is sufficiently large, the surrogate closely matches the hard coverage indicator and the behavior is typically stable. In contrast, $\lambda_{\mathrm{reg}}$ governs the trade-off between the efficiency term $L_{\mathrm{eff}}$ and the coverage term $L_{\mathrm{cov}}$. As stated in our submission, we choose $\lambda_{\mathrm{reg}}$ so that $L_{\mathrm{eff}}$ and $L_{\mathrm{cov}}$ are on comparable scales during training, which helps both objectives contribute meaningfully to optimization. Intuitively, when the intervals are large (and $L_{\mathrm{eff}}$ can dominate), a smaller regularization weight is needed to balance the two terms; conversely, if $\lambda_{\mathrm{reg}}$ is too large, the efficiency component can become negligible relative to $L_{\mathrm{cov}}$ (and effectively disappears), while if it is too small, the coverage term can be overwhelmed and coverage may degrade. We will revise the text to clarify this scaling principle, and we will additionally report the loss trajectories (including $L_{\mathrm{cov}}$ and $L_{\mathrm{eff}}$ separately) to help explain the observed stability and the failure cases such as $\lambda_{\mathrm{reg}}=100$ on Energy.
> > > >
> > > >
> > > > **Q23: Limitations and Drawbacks**
> > > >
> > > > We thank the reviewer for this important suggestion. We agree that a clear discussion of limitations and drawbacks is essential, and we have added an explicit paragraph in Section 6 to address this point. Specifically, we highlight three main limitations of our approach: (1) it relies on held-out data splits for calibration and, when applicable, hyperparameter selection, which can be less effective when data are limited (e.g., sparse EHR measurements). In particular, when the dataset is small, reserving a calibration (and tuning) split reduces the effective sample size for model fitting, which can hurt predictive accuracy and may lead to more conservative (wider) intervals; (2) our main experiments use semi-synthetic missingness, which cannot fully validate performance under the full range of real-world missingness mechanisms; and (3) the current framework is not designed to explicitly handle abrupt regime changes, and incorporating change-point detection or other nonstationarity-aware components is an important direction for future work. Finally, while this work focuses on time series with missing covariates, extending the framework to other sources of non-exchangeability and data imperfections (e.g., label noise or adversarial perturbations) remains an important avenue for future research.

---

> > > > > ### Comment · Reviewer_PgAr · 2026-03-21
> > > > >
> > > > > I thank the authors for their complete and precise responses to all comments and for the substantial revisions to the paper, which have significantly improved the clarity and robustness of the manuscript.

---

> > > > > > ### Author Response · Authors · 2026-04-01
> > > > > >
> > > > > > We thank Reviewer PgAr for their time and for the insightful feedback provided during the review process. We are pleased to hear that our responses and the revisions successfully addressed your concerns. We believe the manuscript is significantly stronger as a result of this change.

---

### Review · Reviewer_Fa7J · 2026-02-02

**Summary Of Contributions:**

**Contributions**

The paper addresses conformal prediction for time series with missing covariates, where structured missingness and temporal dependence make group-conditional coverage difficult to guarantee, and introduces ACP-TSM, a post-training LSTM that learns missingness-aware corrections to nonconformity scores. The three core ideas behind the method are to: (a) leverage, by construction, temporal neighborhood structure; (b) use hand-engineered features to represent structured missingness patterns; and (c) optimize a dual objective that trades off group-conditional coverage gaps across missingness patterns against interval width.

The authors make three main claims: (a) the method improves group-conditional reliability (across MCAR/MAR/MNAR missingness patterns) while producing tighter prediction intervals than prior conformal prediction baselines, (b) the method is theoretically valid in the asymptotic regime, and (c) the method is empirically robust across datasets and imputers.

**Strengths.**

 - *Hard setting.* The paper goes after a legitimately hard setting: non-exchangeable data with arbitrary missingness patterns. The motivation is concrete (data scarcity per mask + distribution shift from both missingness and time dependence), and the proposed method is clearly aimed at those two failure modes.
 - *Clear exposition.* I found the presentation schematic and easy to follow. The method exposition closely follows the empirical intuition laid out in the introduction, which made the design choices feel well motivated rather than arbitrary.
 - *Empirical evidence of better efficiency at similar coverage.* While the gains don't seem uniformly dramatic in every setting, the results generally support the main claim that the method achieves the target coverage while reducing interval width versus baselines.

**Weaknesses.**

  - *Theory is limited in scope.* The main theoretical result provides asymptotic control of coverage error under explicit assumptions (e.g., bounded “true radii”), rather than a finite-sample guarantee of group-conditional validity. Practically, that means the formal result doesn’t fully match the strongest empirical claim (robust conditional coverage across missingness patterns), and most of the heavy lifting is still empirical.
  - *Empirical evaluation is limited.* The empirical evaluation focuses on a limited set of benchmark datasets with predominantly semi-synthetic missingness mechanisms. Additional experiments on datasets with natural missingness, even if restricted to coarse groupings or marginal coverage metrics, would help assess robustness in more realistic settings.

**Additional Comments:**

**Clarifying questions**
- **7**. In Section 4, prediction efficiency (tightness) is described as a goal, but it’s not formally defined. Could you clarify how efficiency is defined, or explicitly link it to the later width/inefficiency objectives in Section 5?
- **8**. In Remark 5.2, is the statement about CP-MDA-Reweighting’s gap growing with more samples meant to be empirical (if so, where is it shown), or theoretical (if so, which result supports it)? Also, Remark 5.2 suggests that using more calibration data yields higher coverage and thus smaller coverage gap, but Theorem 5.1 bounds only time-averaged marginal coverage error (not group coverage gap). Could you please clarify the logical link (or revise this as an empirical/heuristic claim rather than theorem-supported)?
- **9**. Since the method uses a bidirectional LSTM, could you please clarify whether the correction applied at time $t$ depends only on information available up to $t$? In particular: is the BiLSTM trained in a strictly causal (rolling) manner or does it access future calibration points when correcting earlier time steps? At test time, does the interval refinement rely solely on past data, or can future calibration labels influence the correction?

**Audience:**

Yes

**Audience Explanation:**

The problem setting and methodological contributions are likely to be of interest to researchers in uncertainty quantification and time-series modeling.

**Broader Impact Concerns:**

None.

**Claims And Evidence:**

Yes

**Claims Explanation:**

The empirical evidence supports improved efficiency at comparable coverage, but the strongest claims about group-conditional reliability and sample-size effects are not supported by the presented theory and rely largely on semi-synthetic empirical results. Additional empirical evaluation and clarification are therefore requested (see Requested changes below).

_Update after author response: The authors addressed these concerns by adding additional empirical evaluations (including a dataset with natural missingness) and clarifying the theoretical claims. With these revisions, I am satisfied that the claims are now supported._

**Requested Changes:**

**Empirical evaluation**
- **1**. [required] Add at least one evaluation on a dataset with natural missingness. To complement the synthetic missingness injection, include experiments on a dataset where missingness arises organically (e.g., sensor outages, EHR measurements). Even if exact mask-level group-conditional coverage cannot be estimated, reporting marginal coverage and coarse group-conditional metrics would help assess robustness under realistic missingness patterns.
- **2**. [required] Clarify evaluation under distributional shift across time. Since the method leverages temporal neighborhood structure, it would be useful to report performance under regime changes or nonstationary segments (e.g., early vs late periods, high-variance vs stable windows) to better understand when temporal smoothing helps or fails.
- **3**. [required] Expand discussion of limitations tied to semi-synthetic design. Explicitly acknowledge that semi-synthetic missingness enables controlled measurement of group-conditional coverage but may not capture all real-world missingness pathologies.

**Presentation**
- **4**. [optional] I feel the introduction could be clearer about the two main motivations of the work, particularly how missingness complexity and temporal dependence lead to failure modes in existing conformal prediction methods.
- **5**. [optional] The proposed method seems to be much faster than the baselines (Table 20). I feel this could be highlighted more clearly in the main text, possibly by reporting summary on runtimes directly in the text and pointing to Table 20.
- **6**. [required] Expand limitations discussion. What are the main limitations of the method? For example, the approach seems to rely on the assumption that nearby time steps have similar conformal interval behavior—when does this break (e.g., abrupt events, strong nonstationarity)?

---

> ### Author Response · Authors · 2026-02-27
>
> We thank the reviewers for their careful reading of our manuscript and for providing constructive feedback. We have revised the paper accordingly, and all changes are highlighted in blue. Below, we respond to each comment point by point.
>
> **Q1: Evalaution on Dataset with Natural Missingness**
>
> We thank the reviewer for this suggestion. We have added an additional empirical evaluation on a dataset with naturally occurring missingness, AirQualityUCI. This dataset contains 9,358 instances of hourly averaged measurements from an array of five metal-oxide chemical sensors embedded in an Air Quality Chemical Multisensor Device, where missingness arises organically in the collected features rather than being synthetically injected. We remove instances for which the response variable CO(GT) is missing, as well as instances in which all covariates are missing, leaving 7,674 samples for model fitting. Across the covariates, there are 10,583 missing entries in total.
>
> Results are reported in Table 2. Overall, our method attains the desired coverage while producing shorter prediction intervals across scenarios. We do observe notable over-coverage when the number of missing covariates is three, which we attribute to the increased complexity and heterogeneity of naturally occurring missingness. Importantly, even in this setting, our approach remains stable and incurs only lightweight post-hoc overhead, offering a computationally efficient solution in practice. We include the discussion of this natural missingness scenario in Appendix G.12.
>
> **Table 2: Performance on AirQuality (by missingness level and overall)**
>
> | Metric | CP-MDA-Adaptive | MVSC | ACP-TSM |
> |---|---:|---:|---:|
> | *#Missingness = 1* ||||
> | Coverage | 0.865 | 0.913 | 0.891 |
> | Interval width | 3.120 | 4.292 | **2.053** |
> | *#Missingness = 3* ||||
> | Coverage | 0.875 | 0.923 | 0.875 |
> | Interval width | 2.826 | 4.533 | **1.980** |
> | *Overall* ||||
> | Coverage | 0.865 | 0.913 | 0.890 |
> | Interval width | 3.108 | 4.302 | **2.050** |
> | Winkler score | 3.707 | 5.170 | **2.692** |
> | Coverage gap | 0.030 | 0.018 | **0.017** |

---

> > ### Author Response · Authors · 2026-02-27
> >
> > **Q2: Evaluation under Distributional Shift across Time**
> >
> > We thank the reviewer for this suggestion. We empirically verify that our method leverages temporal neighborhood structure on both the oil temperature and electricity datasets. As shown in Figure 6 of our revised manuscript, the oil temperature series exhibits clear seasonal patterns and noticeable distributional shifts over time; nevertheless, the resulting prediction intervals in Figure 2 remain strongly temporally correlated, which motivates our neighborhood-based correction. We agree that an ablation study under regime changes and other nonstationary segments is important. Accordingly, we added two additional evaluations: (1) reporting performance separately on early vs. late time periods, and (2) stratifying test points into three regimes (smooth, intermediate, and abrupt) based on the magnitude of local distributional change, and reporting performance within each regime. For early vs. late periods, as shown in Table 3, the interval performance is relatively stable in the early segment but degrades noticeably in the late segment, consistent with the pronounced distribution shift shown in Figure 6. We believe this is driven by a violation of the exchangeability assumption between the calibration and test points. In particular, even relatively aggressive intervals can maintain coverage guarantee in Seg1, whereas more conservative intervals lead to under-coverage in Seg3. This trend is also visible in Figure~6: Seg1 (1000-2000) appears most aligned with the calibration distribution, and Seg2 (2000-3000) and Seg3 (3000-4000) are more dispersed. Overall, these results underscore the practical importance of the exchangeability assumption that underpins conformal prediction, and highlight robustness to temporal distribution shift as an important direction for real-world deployment.
> >
> > For the  nonstationary segments, we take a closer look by by stratifying time indices according to a simple local change score
> > $d_t = \lvert y_{t-1} + y_{t+1} - 2y_t \rvert,$ and defining three regimes using empirical quantile thresholds (smooth / intermediate / abrupt). We then report coverage and average interval width within each regime in Table 4, which directly diagnoses when temporal smoothing helps or fails under local distributional shift. Surprisingly, performance is broadly consistent across regimes: coverage remains close to the target level in both smooth and abrupt segments, while the intermediate regime exhibits a slight under-coverage; meanwhile, the average width is nearly unchanged across regimes. This indicates that the proposed temporal refinement behaves robustly under local distributional shifts, maintaining similar performance from smooth windows to abrupt changes in this dataset. We provide a more granular discussion of these scenarios in Appendix~G.13.
> >
> > **Table 3: Coverage and interval width across three chronological test segments**
> >
> > | Segment | Coverage | Width |
> > |---|---:|---:|
> > | Seg1 | 0.9110 | 16.7659 |
> > | Seg2 | 0.8650 | 20.2047 |
> > | Seg3 | 0.8340 | 28.4535 |
> >
> >
> > **Table 4: Coverage and interval width by shift regime**
> >
> > |Regime|n|Coverage|Width|
> > |---|---:|---:|---:|
> > |Overall|3000|0.9030|18.6554|
> > |Smooth |1027|0.9049 | 18.6400 |
> > | Intermediate |1008| 0.8978 | 18.6054 |
> > |Abrupt|965|0.9065 | 18.7257 |
> >
> > **Q3 and Q6: Discussion of Limitations.**
> >
> > We thank the reviewer for this important suggestion. We agree that semi-synthetic missingness may not capture the full range of real-world missingness mechanisms and pathologies. Accordingly, we have added an additional experiment on a real dataset with naturally occurring missingness, and we include an explicit paragraph in Section 6 clarifying this point. Specifically, we highlight three main limitations of our method: (1) it relies on a held-out split for model training and hyperparameter tuning, which can be less effective when data are limited (e.g., sparse EHR measurements); (2) our main experiments use semi-synthetic missingness, which cannot validate performance across all real-world missingness mechanisms; and (3) our current framework is not designed to explicitly handle abrupt regime changes; incorporating change-point detection (or other nonstationarity-aware components), as well as extending the method to additional data-imperfection settings (e.g., label noise or adversarial perturbations), is an important direction for future work.
> >
> > **Q4: Introduction and Motivation**
> >
> > We thank the reviewer for highlighting the need to better emphasize the two main motivations of our work. We revised the introduction to more clearly explain (i) how structured missingness can create information-pattern mismatch and lead to undercoverage for missingness-defined subgroups, and (ii) how temporal dependence and nonstationarity violate exchangeability and can degrade calibration over time.

---

> > > ### Author Response · Authors · 2026-02-27
> > >
> > > **Q5: Computational Efficiency and Runtime Analysis**
> > >
> > > We thank the reviewer for this suggestion. We agree that the runtime advantage should be highlighted more clearly. We revised the main text to explicitly summarize the runtime results, emphasizing that the computational gains are most pronounced on the large-scale Oil Temperature and Electricity datasets, and we added a pointer to Table 18 in our revised manuscript, which reports detailed runtimes across methods and datasets. Specifically, on Oil Temperature, our method finishes in approximately 5 minutes, whereas the baselines take over 20 minutes; on Electricity, our method takes about 40 minutes, while the baselines require 10+ hours. This efficiency stems from ACP-TSM’s lightweight post-hoc overhead: it only trains a small LSTM on the calibration set, which is substantially faster than baselines that require additional computation for each subgroup.
> > >
> > > **Q7: Definition for Prediction Efficiency in Section 4**
> > >
> > > We thank the reviewer for pointing this out. We agree that prediction efficiency should be defined
> > > explicitly. We revised Section 3 to define efficiency as the size of the prediction interval: we quantify efficiency by the (empirical) average width $\frac{1}{|K|}\sum_{k\in K}(U_{T+k}-L_{T+k})$, and we use inefficiency interchangeably to refer to this width-based metric (smaller is better).
> > >
> > > **Q8: Clarification on Remark 5.2**
> > >
> > > We thank the reviewer for the careful reading. We agree that Remark~5.2 was potentially misleading, and we have therefore removed it in the revision. Instead, we add a clarification that the theorem establishes an asymptotic guarantee for marginal coverage (i.e., time-averaged miscoverage). We also clarify that (1) the theory relies on a uniform boundedness assumption on the (oracle) radii/interval sizes, which we view as reasonable since achieving the target coverage for any given test point does not require unbounded intervals; and (2) the guarantee is asymptotic as $T\to\infty$, consistent with prior work on adaptive conformal inference.
> > >
> > > **Q9: LSTM Architecture**
> > >
> > > We thank the reviewer for raising this important point. Our procedure is strictly rolling in both training and inference.
> > > The correction applied at time $t$ is computed using only information available up to time $t$ (i.e., past covariates/masks and past nonconformity information), and it never uses future observations or future labels.
> > > Although we implement the correction network as a bidirectional LSTM for convenience, the bidirectionality is applied only within the historical window befor $t$ when forming a representation for time $t$; the input sequence does not include any points after $t$.
> > > At test time, the refined interval for time $t$ depends only on data observed up to $t$; in particular, no future features (or any future labels) are accessed when producing the correction. We also add this explanation in Appendix G.11 in our revised manuscript.

---

> > > > ### Comment · Reviewer_Fa7J · 2026-03-09
> > > >
> > > > Thank you for the revisions. All my questions and requests have been addressed.
> > > >
> > > > For the final version, I’d suggest highlighting the real missingness experiment (Q1) in the main experimental section and including brief pointers to the appendix for the additional analyses under distributional shift (Q2), as well as the Oil Temperature ablation discussed with reviewer 4VNS.

---

> > > > > ### Author Response · Authors · 2026-04-01
> > > > >
> > > > > We thank Reviewer Fa7J for their time and insightful feedback throughout the review process. We are pleased to hear that our responses and revisions have satisfactorily addressed your concerns. In the revised manuscript, we will move Appendix G.12, which analyzes natural missingness, and Appendix G.3, which presents the ablation study, into the main text. We will also incorporate the analysis of distribution shift from our response to Q2 (currently in Appendix G.13) into the main-text sensitivity analysis.

---

### Review · Reviewer_4VNS · 2026-02-14

**Summary Of Contributions:**

The paper introduces ACP-TSM, a post-hoc conformal prediction framework for time series with missing covariates that aims to achieve both marginal and group-conditional coverage while maintaining tight prediction intervals. The key idea is to train a lightweight LSTM on calibration data to adjust conformal nonconformity scores, optimizing a dual objective that penalizes worst-group coverage gaps and interval width. The method is model-agnostic, imputation-agnostic, and computationally efficient compared to prior approaches that rely on matching missingness patterns or adaptive weighting. Empirically, it achieves reliable coverage with improved efficiency across MCAR, MAR, and MNAR settings. The method has a clear motivation, practical design, and strong empirical results across multiple datasets and missingness mechanisms.

**Audience:**

No

**Audience Explanation:**

The paper addresses conformal prediction under non-exchangeability and structured missingness in time series -- an increasingly relevant setting in applied ML domains such as healthcare, energy, and finance. Researchers working on uncertainty quantification, conformal prediction, time-series modeling, and robustness to missing data would likely find the methodological contributions and empirical findings informative.

**Claims And Evidence:**

Yes

**Claims Explanation:**

The empirical evaluation is extensive and systematically compares the proposed method against strong baselines across multiple datasets and missingness mechanisms (MCAR, MAR, MNAR), reporting marginal coverage, group-conditional coverage, interval width, and Winkler scores. The results consistently show that ACP-TSM achieves target coverage while producing tighter intervals, and ablation studies support the necessity of the proposed components. While some guarantees are asymptotic and experiments rely partly on controlled missingness scenarios, the overall evidence is clear and thorough.

**Requested Changes:**

[high]

* The paper claims improved group-conditional coverage and provides an asymptotic bound, but the exact scope and limitations of these guarantees (e.g., dependence on initialization, smooth surrogate coverage loss) should be stated more explicitly in the main text. In particular, clarify what is formally guaranteed (marginal vs. group-conditional coverage) and under which assumptions.

* Provide a clearer quantitative breakdown of how each component -- coverage loss, inefficiency loss, adaptive \alpha-adjustment, and group features -- affects marginal coverage, group coverage gap, and interval width. A step-by-step ablation (adding one component at a time) would help clarify their individual and combined contributions.

[medium-low]

* While MNAR scenarios are simulated, including experiments with naturally occurring missingness (if available) would improve practical relevance.

* Expand discussion of hyperparameter sensitivity (\lambda_reg, smoothing factor c) and training stability of the post-hoc LSTM.

* Include clearer reporting of runtime settings and complexity scaling with dimensionality and number of missing patterns.

* Clarify the practical deployment protocol, including how often recalibration or LSTM retraining is required under distribution shift, and whether the correction model must be updated online, to better understand how the method would be maintained in real-world time-series applications.

---

> ### Author Response · Authors · 2026-02-27
>
> We thank the reviewers for their careful reading of our manuscript and for providing constructive feedback. We have revised the paper accordingly, and all changes are highlighted in blue. Below, we respond to each comment point by point.
>
> **Q1: Clarifying Coverage Guarantees**
>
> We thank the reviewer for raising this important point. We added a clarification at the end of Section~4 to more explicitly state the scope, assumptions, and limitations of our theoretical guarantees. In particular, our formal result provides an asymptotic guarantee for marginal coverage (time-averaged miscoverage), rather than group-conditional coverage; improvements in group-conditional coverage are supported empirically in our experiments. We also clarify that (1) the theory relies on a uniform boundedness assumption on the (oracle) radii / interval sizes, which we view as reasonable since achieving the target coverage for any test point does not require unbounded intervals; and (2) the guarantee is asymptotic as $T\to\infty$, consistent with prior work in adaptive conformal inference.
>
> **Q2: Step by step ablation of key components**
>
> We thank the reviewer for this valuable suggestion. In response, we conduct a step-by-step ablation study of the key components on the Oil Temperature dataset under both MCAR and MAR missingness in Table 1. Adding the coverage loss consistently moves empirical coverage closer to the target level, while adding the inefficiency loss improves efficiency by reducing interval width. Although there are minor fluctuations across variants, the adaptive fine-tuning module and the incorporation of group features further stabilize performance and yield the most reliable overall results across both missingness settings. We've included these additional results in Appendix G.3 in our revised manuscript.
>
> **Table 1: Step-by-step ablation on the Oil Temperature dataset (MCAR & MAR)**
>
> | Setting | Variant | Marginal coverage (%) | Avg. coverage gap (%) | Interval width |
> |---|---|---:|---:|---:|
> | *MCAR* | B0: Base | 0.812 | 0.101 | 21.78 |
> | *MCAR* | B1: + coverage loss | 0.870 | 0.035 | 25.71 |
> | *MCAR* | B2: + inefficiency loss | 0.859 | 0.043 | 25.26 |
> | *MCAR* | B3: + adaptive α adjustment | 0.892 | 0.009 | 25.68 |
> | *MCAR* | B4: + group features (Full) | 0.896 | 0.001 | 25.70 |
> | *MAR* | B0: Base | 0.809 | 0.101 | 22.72 |
> | *MAR* | B1: + coverage loss | 0.861 | 0.041 | 25.39 |
> | *MAR* | B2: + inefficiency loss | 0.862 | 0.042 | 25.38 |
> | *MAR* | B3: + adaptive α adjustment | 0.908 | 0.010 | 25.84 |
> | *MAR* | B4: + group features (Full) | 0.901 | 0.008 | 25.79 |
>
> **Q3: Evaluation on Dataset with Natural Missingness**
>
> We thank the reviewer for this suggestion. We have added an additional empirical evaluation on a dataset with naturally occurring missingness, AirQualityUCI. This dataset contains 9,358 instances of hourly averaged measurements from an array of five metal-oxide chemical sensors embedded in an Air Quality Chemical Multisensor Device, where missingness arises organically in the collected features rather than being synthetically injected. We remove instances for which the response variable CO(GT) is missing, as well as instances in which all covariates are missing, leaving 7,674 samples for model fitting. Across the covariates, there are 10,583 missing entries in total.
>
> Results are reported in Table 2. Overall, our method attains the desired coverage while producing shorter prediction intervals across scenarios. We do observe notable over-coverage when the number of missing covariates is three, which we attribute to the increased complexity and heterogeneity of naturally occurring missingness. Importantly, even in this setting, our approach remains stable and incurs only lightweight post-hoc overhead, offering a computationally efficient solution in practice. This additional result is included in Appendix G.12 in our revised manuscript.
>
> **Table 2: Performance on AirQuality (by missingness level and overall)**
>
> | Metric | CP-MDA-Adaptive | MVSC | ACP-TSM |
> |---|---:|---:|---:|
> | *#Missingness = 1* ||||
> | Coverage | 0.865 | 0.913 | 0.891 |
> | Interval width | 3.120 | 4.292 | **2.053** |
> | *#Missingness = 3* ||||
> | Coverage | 0.875 | 0.923 | 0.875 |
> | Interval width | 2.826 | 4.533 | **1.980** |
> | *Overall* ||||
> | Coverage | 0.865 | 0.913 | 0.890 |
> | Interval width | 3.108 | 4.302 | **2.050** |
> | Winkler score | 3.707 | 5.170 | **2.692** |
> | Coverage gap | 0.030 | 0.018 | **0.017** |

---

> > ### Author Response · Authors · 2026-02-27
> >
> > **Q4: Expand discussion of hyperparameter sensitivity and training stability of the post-hoc LSTM.**
> >
> > We thank the reviewer for this helpful suggestion. In the revised manuscript, we expand the discussion of hyperparameter sensitivity and training stability of the post-hoc LSTM.
> > First, we explain that $\lambda_{\mathrm{reg}}$ is the weighting parameter that balances our two loss components, namely the conditional coverage loss and the inefficiency (interval width) loss. We add a more detailed description of how extreme choices of $\lambda_{\mathrm{reg}}$ affect the optimization: overly large $\lambda_{\mathrm{reg}}$ can over-emphasize efficiency and shrink intervals aggressively, potentially slowing the reduction of coverage gap, whereas overly small $\lambda_{\mathrm{reg}}$ can make the coverage term dominate and lead to overly conservative (wide) intervals.
> >
> > Second, we explicitly describe $c$ as a smoothing factor used to construct a differentiable surrogate for the coverage-related loss. We add discussion on the tradeoff induced by $c$: smaller $c$ yields a sharper surrogate that is closer to the original non-smooth objective but may increase gradient variance, while larger $c$ produces a smoother surrogate that improves numerical stability but can weaken the corrective signal near constraint violations. We emphasize that our results are stable across a broad range of $c$ in Table~6.
> >
> > Third, motivated by our empirical finding that the scale of the coverage loss is relatively stable across datasets while the inefficiency loss can vary substantially, we add an automatic scaling heuristic to guide $\lambda_{\mathrm{reg}}$ selection. Concretely, we describe and empirically validate a simple rule that chooses $\lambda_{\mathrm{reg}}$ to match the magnitudes of the two loss terms, and we report that the product $\lambda_{\mathrm{reg}}\times(\text{typical interval width})$ consistently falls in the range of $10$--$100$ across datasets. We further include an expanded hyperparameter search on the Energy dataset in Table~7 in our revision to confirm that performance remains stable when this scale-matching criterion is satisfied.
> >
> > Finally, since the LSTM is trained post-hoc after the base predictor is fixed, it serves as a lightweight calibrator rather than a jointly-trained forecaster, which significantly improves stability.
> >
> > **Q5: Runtime analysis and complexity scaling with dimensionality and number of missing patterns.**
> >
> > We thank the reviewer for this helpful suggestion. In the revised manuscript, we revised our runtime analysis in Table 18 in our manuscript to explicitly report the covariate dimensionality for each dataset: Electricity has 4 covariates, Oil Temperature has 6 covariates, and the Energy dataset has 27 covariates. We further clarify the computational implications of increasing dimensionality. Methods such as CP-MDA-Adaptive and MVSC rely on group-level calibration defined by missingness patterns. As the dimensionality increases, the number of possible missingness-pattern groups grows exponentially, which substantially increases computational cost and can limit practical applicability. In contrast, our method calibrates using the full calibration set without enumerating missingness-pattern groups, so its computational cost scales approximately linearly with dimensionality, making it more suitable for higher-dimensional settings such as the Energy dataset.
> >
> > **Q6: Practical Deployment and Model Maintenance Under Distribution Shift**
> >
> > We thank the reviewer for raising this important point. We agree that, in principle, periodically retraining the correction model can help incorporate the most recent information and improve performance under distribution shift in time series settings.
> > On the other hand, our empirical results suggest that frequent LSTM retraining is not always necessary. For the Oil Temperature dataset, Figure 6 in the manuscript shows a substantial shift during the test period, yet the performance of our method remains stable. We attribute this robustness primarily to the adaptive fine tuning component, which continuously leverages recent observations to adjust the prediction procedure. Results in Table 14 in our manuscript further suggest that our method can successfully handle multi-step forecasting.
> >
> > Based on these findings, we view a practical deployment protocol as follows: the base LSTM can be retrained only occasionally (for example, when sufficient new data accumulate or when predictive accuracy degrades). We agree, however, that this involves a tradeoff between empirical performance and computational cost, and we clarify this maintenance protocol and the associated tradeoffs in Appendix H in our revised manuscript.

---

> > > ### Comment · Reviewer_4VNS · 2026-04-03
> > >
> > > I thank the authors for their thorough and precise responses to all comments, as well as for the substantial revisions made to the paper, which have greatly enhanced the manuscript’s clarity.

---

### Decision · Action_Editor_bbir · 2026-04-13

**Recommendation:** Accept as is

**Audience:**

Yes

**Audience Explanation:**

All reviewers are aligned that the audience is a match.

**Claims And Evidence:**

Yes

**Claims Explanation:**

All three reviewers provided in-depth reviews that the authors responded to; the reviewers are happy with the author response and the edits made to the manuscript.